# LEVERAGING DISCRETE FUNCTION DECOMPOSABILITY FOR SCIENTIFIC DESIGN

**James C. Bowden**[1], **Sergey Levine**[1], **Jennifer Listgarten**[1,2,3]
{jcbowden, sergey.levine, jennl}@berkeley.edu

## ABSTRACT

In the era of AI-driven science and engineering, we often want to design discrete objects (*e.g.*, circuits, proteins, materials) *in silico* according to user-specified properties (*e.g.*, that a protein binds its target). Given a property predictive model, *in silico* design typically involves training a generative model over the design space (*e.g.*, over the set of all length-$L$ proteins) to concentrate on designs with the desired properties. *Distributional optimization*, formalized as an estimation of distribution algorithm or as reinforcement learning policy optimization, maximizes an objective function in expectation over samples. Optimizing a distribution over discrete-valued designs is in general challenging due to the combinatorial nature of the design space. However, many property predictors in scientific applications are *decomposable* in the sense that they can be factorized over design variables in a way that will prove useful. For example, the active site amino acids in a catalytic protein may need to only loosely interact with the rest of the protein for maximal catalytic activity. Current distributional optimization algorithms are unable to make use of such structure, which could dramatically improve the optimization. Herein, we propose and demonstrate use of a new distributional optimization algorithm, DECOMPOSITION-AWARE DISTRIBUTIONAL OPTIMIZATION (DADO), that can leverage any decomposability defined by a junction tree on the design variables. At its core, DADO employs a factorized "search distribution"—a learned generative model—for efficient navigation of the search space, and invokes graph message passing to coordinate optimization across all variables.

## 1 DESIGN IN DISCRETE STATE SPACES

The integration of AI into scientific research has opened new avenues for property-driven, *in silico* design of discrete objects—from molecular structures like proteins, to fabricated systems like circuits—where computational methods guide design with user-specified properties. For example, we may seek to design an amino acid sequence for a protein so that the protein binds its target.

Given a property predictive model, $f(x)$, the simplest version of *in silico* design entails enumerating all possible designs, $x = [x_1, x_2, ..., x_L] \in \mathcal{X}$ (*e.g.*, all possible amino acid sequences of length $L$), evaluating each one under the predictive model, $s = f(x)$, and choosing the design with the highest $s$. Such a setup is complicated by two primary challenges. First, in most realistic problems, the design space is too large to fully enumerate. Second, if $f(x)$ has parameters estimated from data, then it is most likely not accurate over the whole space. While many works address the second problem, there has been little recent development on the first; as such, we focus herein on performing efficient optimization in high-dimensional discrete design spaces. The second problem can be handled with approaches complementary to ours (*e.g.*, Brookes et al. (2019); Trabucco et al. (2021); Uehara et al. (2024)), and is not here considered. We will focus our examples and experiments on designing amino acid sequences, but our method is general—applicable to design on any discrete space.

Finding the highest-scoring design, $x^*$, naively requires $D^L$ evaluations of $f(x)$ for a $D$-amino acid alphabet (typically $D = 20$). To develop some intuition, in this section only, let us assume that $D = 2$ so that $x$ is a binary sequence. First, consider the simple but unrealistic case where $f(x) = \beta_1 x_1 + \beta_2 x_2, \ldots, \beta_L x_L$, for scalar parameters $\{\beta_i\}$. In such a scenario, finding the design

---

[1]Department of Electrical Engineering and Computer Sciences, UC Berkeley. [2]Center for Computational Biology, UC Berkeley. [3]UC Berkeley–UCSF Graduate Program in Bioengineering, UC San Francisco.

with the highest $f(x)$ requires considering only $D \times L$ partial designs because the $L$ components of $x$ do not interact with each other, acting only linearly additively. A more realistic setting would allow for more complicated functional forms while maintaining some notion of linear additivity. For example, consider the form $f(x) = C_1(\hat{x}_1) + C_2(\hat{x}_2), \ldots, C_\kappa(\hat{x}_\kappa)$, where $C_i$ denotes an arbitrary function on a set of design variables, $\hat{x}_i$, such as $C_1(\hat{x}_1) = C_1(x_1, x_3, x_8) = \exp(7x_1x_3 - 3x_3x_8)$. In the case where sets of variables do not overlap, *i.e.*, where $x_i$ can appear in only one component $C_j$, finding the global optimum of $f(x)$ requires a number of evaluations of component functions that scales as $D^M$, where $M \leq L$ is the cardinality of the largest variable set. When the sets of variables *do* overlap across component functions, tying them together, then the number of required evaluations becomes correspondingly higher. Note that this more general form of linear additivity admits representation of any function, possibly requiring that $M = L$ in which case there is no linear additivity, nor consequently, decomposability. In realistic settings with many design variables and a larger alphabet (*e.g.*, $D = 20$), the difference in number of evaluations required between the decomposed and standard scenarios is even greater still. Importantly, in most real problems, we expect some level of decomposability. For example, in designing a protein to bind to a target, a few key sequence positions may make up the binding interface, which primarily dictates the binding strength, whereas the other positions may be involved in stabilizing the protein for binding.

Classical message passing algorithms can leverage the aforementioned general type of structure in $f(x)$ to exactly find the global optimizer with the lowest possible time complexity (Vlassis et al., 2004). However, for reasons discussed momentarily, we are interested in *distributional optimization*, wherein a standard optimization problem over the design space, $x^* = \arg\max_{x \in \mathcal{X}} f(x)$, is replaced by one over the parameters of a generative model, $p_\theta(x)$, namely, $\theta^* = \arg\max_\theta \mathbb{E}_{p_\theta(x)}[f(x)]$. These formulations are equivalent in that the optimum of each is the same, assuming that $p_\theta(x)$ has the capacity to place all of its mass on $x^*$. However, each formulation lends itself to different algorithms and extensions, mentioned below. Distributional optimization may employ strategies to prevent $p_\theta(x)$—the search distribution/policy—from collapsing to a point mass, such as by using a prior (Brookes et al., 2019) or entropy regularizer (Ziebart et al., 2008).

Our interest in the distributional optimization formulation is motivated by its extensibility. First, such a setup enables us to directly use innovations from the Estimation of Distribution Algorithm (EDA) (Brookes et al., 2020; Larrañaga & Lozano, 2001) and policy optimization (Peters & Schaal, 2007; Peng et al., 2019) communities. Of particular note are methods that enable combining a pre-trained, unconditional generative model, $p(x)$, with a property predictor, $p(y|x)$, to execute Bayes rule and thereby obtain a sampling distribution that can be used for design, $p(x|y \in Y)$ (*e.g.*, Brookes et al. (2019); Fannjiang & Listgarten (2020); Uehara et al. (2024)). Second, as the key object that navigates the search space, $p_\theta(x)$, is a generative model, we stand to benefit from advances in generative modeling. Herein, for clarity of contribution, we focus on the purest form of the EDA, without entropy regularization or a prior. We leave such extensions to future work.

**Contributions.** We develop a distributional optimization algorithm in the form of a generalized EDA/policy optimization algorithm that can leverage any decomposability in $f(x)$ in the form described earlier as $f(x) = C_1(\hat{x}_1) + C_2(\hat{x}_2), \ldots, C_\kappa(\hat{x}_\kappa)$, to more efficiently navigate the design space and find high-performing designs quickly (Fig. 1). We call our method DECOMPOSITION-AWARE DISTRIBUTIONAL OPTIMIZATION (DADO). We first empirically investigate DADO on synthetic examples, illustrating that the anticipated optimization efficiency emerges compared to decomposition-unaware baselines. Then we further substantiate this efficiency on problems anchored on real protein data, that is, by optimizing protein property predictive models. Additionally, in case studies on a few protein predictive models, we find that accuracy is robust to modifications of the decomposition, suggesting that perfect *a priori* knowledge of decomposability is not required.

As DADO requires a decomposed objective of the form $f(x) = C_1(\hat{x}_1) + C_2(\hat{x}_2), \ldots, C_\kappa(\hat{x}_\kappa)$, on our protein data problems, we construct these as follows. We first obtain a graph topology of which variables (protein residues) are coupled by thresholding residue distances from an AlphaFold3 structure (Abramson et al., 2024; Brookes et al., 2022). This graph dictates the functional form of the predictive model, by way of an automatically constructed junction tree. Recall that junction trees represent arbitrarily complex relationships between random variables by transforming any graph into a tree structure, where each node contains a set of the original variables, and where these sets may be overlapping in adjacent nodes (Lauritzen & Spiegelhalter, 1988). Then we fit a neural network that enforces this decomposition, to the training data. Analogous procedures in different application areas could include the following, for example. In circuit design, certain topologies of components

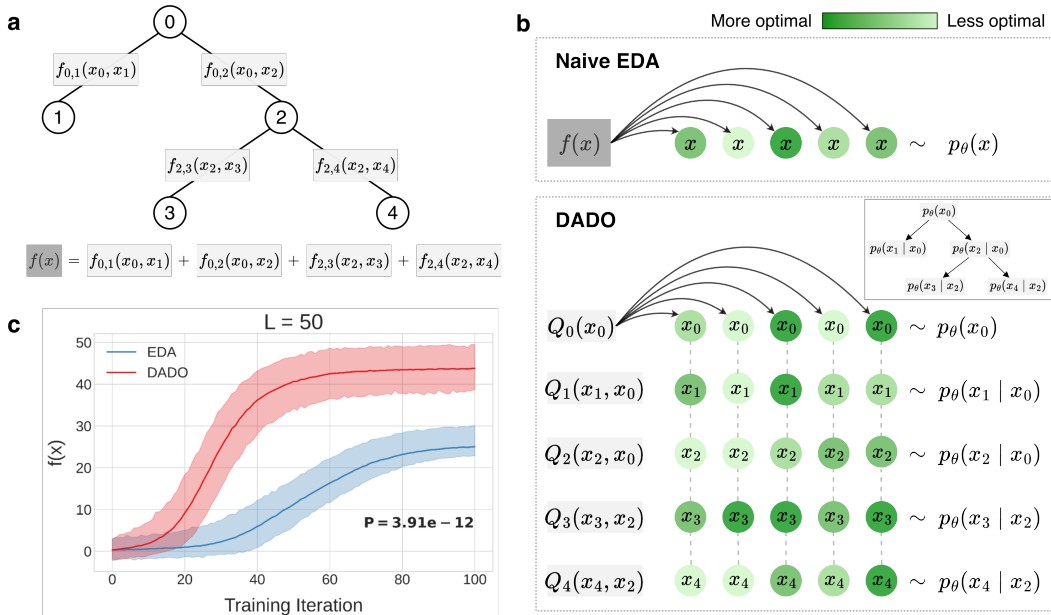

**Figure 1: Key components of DADO. a,** DADO requires as input an objective function in its decomposed form, $f(x) = C_1(\hat{x}_1) + C_2(\hat{x}_2), \dots, C_\kappa(\hat{x}_\kappa)$, which corresponds to a junction tree. Here we show a junction tree with nodes of size 1, *i.e.*, a regular tree, for simplicity. Variables with edges interact to directly influence $f$. Some variables participate in multiple component functions, requiring coordination. **b,** To update the search distribution at each iteration, naive EDAs weight entire samples, drawn from a joint distribution over all design variables, with $f(x)$. In contrast, DADO leverages the decomposition of $f(x)$ to separately weight sample dimensions for each node. DADO uses message-passing to compute *value functions*, $Q_i$, accounting for each $x_i$ interacting with its descendants. These $Q_i$ serve as weights for each part of the search distribution, which is factorized like $f$. **c,** Example performance comparison on a synthetic problem with an exact tree decomposition over a discrete design space of size $20^{50}$ ($D = 20, L = 50$). Each method drew 100 samples per iteration. We evaluated these samples with $f(x)$, computing the per-iteration mean and 95% confidence interval. Results shown were averaged over 20 random seeds for the same $f$ (details in Sec. 4). The p-value is from a two-sided paired t-test that the mean at the final iteration is different between methods, over the 20 seeds.

may be prescribed ahead of time according to production constraints or domain knowledge. When designing a telescope, designers consider different arrangements of lenses—*i.e.*, topologies—and optimize the diameter, curvature, material, coating, etc. for each component.

## 2 DECOMPOSITION-AWARE DISCRETE OPTIMIZATION (DADO)

We begin by reminding the reader of the three primary steps that are iterated in a standard EDA, which is unaware of any decomposability (*e.g.*, Brookes et al. (2020)). From there, we describe how infusing this algorithm with awareness of the known decomposition of $f(x)$ stands to make the EDA more statistically efficient, after which we formally introduce DADO.

Intuitively, one can think of the search distribution as a spotlight on the design space, which iteratively gets moved toward areas with high $f(x)$ in expectation. Modern EDAs parameterize the search model with neural network generative models, such as a Variational Autoencoder (*e.g.*, Brookes et al. (2019)). In a standard EDA, after initializing the search distribution, $p_{\theta^0}(x)$, the EDA proceeds by iterating through these three steps either $N$ times or until convergence (Sec. A.2):

1. Draw $K$ samples from the current search distribution, $\{x^k\}_{k=1}^K \sim p_{\theta^n}(x)$
2. Score each sample with the objective function to get its weight, $w^k = f(x^k)$.
3. Update search distribution parameters by weighted maximum likelihood estimation (MLE), using the weighted samples, $\theta^{n+1} = \arg\max_\theta \mathbb{E}_{\{x^k\}} w^k \log p_\theta(x^k)$.

The weighted MLE problem is typically solved with gradient descent; choosing to use only a fixed number of gradient steps, as we will do, can be theoretically justified through the equivalence of EDAs to Expectation-Maximization (Brookes et al., 2020).

Now that a standard EDA is fresh in our minds, we can consider what it might mean to infuse it with knowledge of how $f(x)$ is decomposed, and how this might prove useful. Before discussing the general case, let us first build intuition with a simpler example. Specifically, consider $f(x) = C_1(\hat{x}_1) + C_2(\hat{x}_2)$ where there is no overlap in variables, $x_i$, between the meta-variables $\hat{x}_1$ and $\hat{x}_2$. In this setting, we can replace step 1, of sampling from one search model over all the design variables, $x = [\hat{x}_1, \hat{x}_2] = [x_1, x_2, ..., x_L] \sim p_{\theta^n}(x)$, with instead sampling each meta-variable from its own search model separately, $\hat{x}_1^k \sim p_{\theta_1^n}(\hat{x}_1)$ and $\hat{x}_2^k \sim p_{\theta_2^n}(\hat{x}_2)$. Having done so, we can execute steps 2 and 3 in a similar manner by scoring each meta-variable separately with its component function ($e.g.$, $w_1^k = C_1(\hat{x}_1^k)$) and then updating each search model independently of the other, via weighted MLE, to obtain $p_{\theta_1^{n+1}}(\hat{x}_1)$ and $p_{\theta_2^{n+1}}(\hat{x}_2)$ (see Sec. A.3 for side-by-side algorithm sketches).

What has this bought us? We have split one optimization problem over a combinatorial space into two independent optimization problems, each over a much smaller space. For a problem where each meta-variable comprises just 15 binary design variables, we have transformed an optimization over a space of size $2^{30} \approx 10^9$ to two problems each of size $2^{15} \approx 10^5$. For larger alphabets and sequence lengths, these differences will be larger still. Importantly, the weighted MLE update with a fixed number of samples, $K$, is more statistically efficient for each lower-dimensional search model than it is for a decomposition-unaware EDA using a single joint search model over all design variables.

The general case allows overlapping variables across component functions, requiring coordination between meta-variables—we cannot divide and conquer as above; rather, we must divide, collaborate, and then conquer. It will be necessary to equivalently describe the decomposability of $f(x)$ with a junction tree, defined by nodes corresponding to meta-variables and undirected edges connecting them (details in Sec. 2.1 and Sec. A.1). Given such a junction tree decomposition, we use a factorized search distribution—a directed, acyclic graphical (DAG) model—matching its topology. Each factor specifies the density of a node's variables conditioned on its parent node's variables, and will be parameterized by a neural network such as an autoregressive model. Sampling from such a distribution is both computationally efficient and easy, so step 1 of the EDA remains straightforward. However, steps 2 and 3 of the standard EDA are not so easily generalized. To update the factorized search distribution in a manner that reduces the effective size of the optimization problem, we will need to generalize "max-plus" message passing, which yields a global optimum $x^* = \arg\max_{x \in \mathcal{X}} f(x)$ (Vlassis et al., 2004), to a procedure integrating message passing with a factorized search model fitting step, thus constituting a decomposition-aware EDA. Having provided an intuitive overview of DADO (Fig. 1), we now proceed to a more formal exposition.

## 2.1 Formal Exposition of DADO

The goal of DADO is to obtain a generative model, $p_\theta(x)$, that maximizes the EDA objective, $\arg\max_\theta \mathbb{E}_{p_\theta(x)}[f(x)]$, while leveraging decomposability in $f(x)$ for optimization efficiency. DADO requires as input a decomposed version of $f(x)$, which can be described by an undirected junction tree, $\mathcal{T} := (\mathcal{N}, \mathcal{E})$, with nodes, $\mathcal{N}$, and edges, $\mathcal{E}$. As noted in the introduction and shown in our experiments on proteins, identifying useful decomposability is feasible in practice. Given the junction tree topology, we write $f(x) = \sum_{i \in \mathcal{N}} f_i(\tilde{x}_i) + \sum_{(i,j) \in \mathcal{E}} f_{i,j}(\tilde{x}_i, \tilde{x}_j)$, where we refer to $f_i(\tilde{x}_i)$ and $f_{i,j}(\tilde{x}_i, \tilde{x}_j)$ respectively as node and edge component functions (Fig. 1a); these are intimately related to the "epistatic landscape" of a protein property function (Sec. A.1.1). When the component functions are not known *a priori*, they can be parameterized and fit to labeled data.

We will begin by recalling how to do decomposition-aware exact (non-distributional) optimization, that is, to solve $\arg\max_x f(x)$. This problem is efficiently solved with a classical message-passing algorithm, which coordinates local optimizations across parts of the junction tree to obtain a single global optimum. Its efficiency comes from breaking optimization over all variables jointly into separate optimizations for each (smaller) meta-variable. Having loaded the reader with this intuition, we then adapt these ideas to distributional optimization, yielding DADO.

### 2.1.1 Classical message-passing for non-distributional optimization

Although classical message-passing has been largely used for probabilistic inference on probabilistic graphical models (Pearl, 1988; Shah, 2014), it can also be used for exact optimization of a function, $x^* = \arg\max_x f(x)$ (Vlassis et al., 2004). Given a function defined on an undirected junction tree, $\mathcal{T}$, message-passing makes use of its topology to find a global maximum. In particular, one roots

the undirected junction tree to obtain a directed tree, inducing a hierarchy among the meta-variables from root to leaves. Each node is responsible for accumulating information from all nodes in its sub-tree and then passing this information on to its parent. Consequently, the root node receives information from the entire tree, which is sufficient to set its variables in a globally optimal manner. Then, starting with the root, each parent communicates its variables' optimal settings to its children, which can in turn set their variables optimally, and so forth.

To obtain the rooted tree, $\mathcal{T}' := (\mathcal{N}', \mathcal{E}')$, from the junction tree, one keeps the same nodes, $\mathcal{N}' = \mathcal{N}$, chooses a root node, $r$, and directs all edges in $\mathcal{E}$ outward from $r$, yielding directed edges, $\mathcal{E}'$. Although rooting at any node will suffice, we choose $r$ such that $\mathcal{T}'$ has the shortest height possible. Message-passing finds a global optimum in two passes through the tree: one round of passing messages up from leaves to root and one round passing messages back down to the leaves.

Given $\mathcal{T}'$, classical message-passing first uses dynamic programming to accumulate information from the leaves up to the root. Similarly to any dynamic programming procedure, we accumulate solutions to increasingly larger intermediate sub-problems. In this case, each sub-problem is to find the value of $f(x)$ evaluated on only a subset of meta-variables, rather than on the full set of variables in $x$. Each sub-problem is tractable owing to the decomposition of the objective function into component functions for each node and edge, and by respecting the partial order of sub-problems induced by $\mathcal{T}'$. Specifically, one computes a *value function*, $V_i^{\max}(\tilde{x}_{\mathrm{p}(i)})$, for each node $i \in \mathcal{N}' \setminus \{r\}$, which tells us for each setting of its parent, $\tilde{x}_{\mathrm{p}(i)}$, the value of the intermediate objective function defined by the edge component function, $f_{\mathrm{p}(i),i}(\tilde{x}_{\mathrm{p}(i)}, \tilde{x}_i)$, plus all component functions over the sub-tree rooted at $i$, given that all nodes maximize their respective intermediate objectives. Computing value functions constitutes the first pass through the tree, from leaves to root:

$$V_i^{\max}(\tilde{x}_{\mathrm{p}(i)}) := \max_{\tilde{x}_i} \left( f_i(\tilde{x}_i) + f_{\mathrm{p}(i),i}(\tilde{x}_{\mathrm{p}(i)}, \tilde{x}_i) + \sum_{c \in \mathrm{children}(i)} V_c^{\max}(\tilde{x}_i) \right).$$

Notably, $V_i^{\max}(\tilde{x}_{\mathrm{p}(i)})$ provides sufficient information about all nodes in the sub-tree rooted at $i$ to optimally choose the value of $\tilde{x}_{\mathrm{p}(i)}$ with respect to its children. Thus it follows that once all value functions have been computed, the root's assignment can be set in a globally optimal manner from its children's value functions,

$$\tilde{x}_r^* := \arg\max_{\tilde{x}_r} \left( f_r(\tilde{x}_r) + \sum_{c \in \mathrm{children}(r)} V_c(\tilde{x}_r) \right).$$

Having chosen the root assignment, we then pass it down the tree as $\tilde{x}_{\mathrm{p}(i)} = \tilde{x}_{\mathrm{p}(i)}^*$ to its children, which successively pass their chosen assignments to their children, all the way to the leaves,

$$\tilde{x}_i^* := \arg\max_{\tilde{x}_i} \left( f_i(\tilde{x}_i) + f_{\mathrm{p}(i),i}(\tilde{x}_{\mathrm{p}(i)}^*, \tilde{x}_i) + \sum_{c \in \mathrm{children}(i)} V_c(\tilde{x}_i) \right),$$

resulting in a global maximizer $x^*$ of $f(x)$. This dynamic programming "traceback" of optimal assignments back down the tree constitutes our second and final pass of messages.

**Alternative notation.** For convenience of our generalization to distributional optimization, we re-write the parent value functions $V_i^{\max}(\tilde{x}_{\mathrm{p}(i)})$ in terms of child-parent, $Q_i^{\max}(\tilde{x}_i, \tilde{x}_{\mathrm{p}(i)})$, and single-node, $Q_i^{\max}(\tilde{x}_i)$ value functions:

$$V_i^{\max}(\tilde{x}_{\mathrm{p}(i)}) := \max_{\tilde{x}_i} Q_i^{\max}(\tilde{x}_i, \tilde{x}_{\mathrm{p}(i)}), \quad \text{where} \tag{1}$$

$$Q_i^{\max}(\tilde{x}_i, \tilde{x}_{\mathrm{p}(i)}) := f_{\mathrm{p}(i),i}(\tilde{x}_{\mathrm{p}(i)}, \tilde{x}_i) + Q_i^{\max}(\tilde{x}_i) \text{ and } Q_i^{\max}(\tilde{x}_i) := f_i(\tilde{x}_i) + \sum_{c \in \mathrm{children}(i)} V_c^{\max}(\tilde{x}_i).$$

In contrast to the original parent value functions, $Q_i^{\max}(\tilde{x}_i, \tilde{x}_{\mathrm{p}(i)})$ represents the effect of the choice of *both* $\tilde{x}_{\mathrm{p}(i)}$ and $\tilde{x}_i$ on their edge component function plus all component functions over the sub-tree rooted at $i$, assuming all descendants of $i$ maximize their corresponding value functions. Intuitively, $Q_i^{\max}(\tilde{x}_i, \tilde{x}_{\mathrm{p}(i)})$ is the value function on edge $(\mathrm{p}(i), i)$ prior to $\tilde{x}_i$ being maximized out, which will become useful if we want to, say, sample $\tilde{x}_i$ according to some distribution instead. $Q_i^{\max}(\tilde{x}_i, \tilde{x}_{\mathrm{p}(i)})$ is composed of two terms, one of which depends on its parent, and one of which doesn't, $Q_i^{\max}(\tilde{x}_i)$. Written using the $Q$-functions just defined, the equivalent traceback equations for selecting a globally optimal assignment, $x^*$, are simply $\tilde{x}_r^* := \arg\max_{\tilde{x}_r} Q_r^{\max}(\tilde{x}_r)$ and $\tilde{x}_i^* := \arg\max_{\tilde{x}_i} Q_i^{\max}(\tilde{x}_i, \tilde{x}_{\mathrm{p}(i)}^*)$. In other words,

$$x^* = \arg\max_x f(x) = \{\arg\max_{\tilde{x}_r} Q_r^{\max}(\tilde{x}_r)\} + \{\arg\max_{\tilde{x}_i} Q_i^{\max}(\tilde{x}_i, \tilde{x}_{\mathrm{p}(i)}^*)\}_{(\mathrm{p}(i),i) \in \mathcal{E}'} \tag{2}$$

$$= \arg\max_x \left( Q_r^{\max}(\tilde{x}_r) + \sum_{(\mathrm{p}(i),i) \in \mathcal{E}'} Q_i^{\max}(\tilde{x}_i, \tilde{x}_{\mathrm{p}(i)}) \right). \tag{3}$$

### 2.1.2 FROM CLASSICAL MESSAGE PASSING TO DISTRIBUTIONAL OPTIMIZATION

In the same way that an EDA transforms $\arg\max_x f(x)$ into a distributional optimization problem, we can rewrite the equivalent message-passing objective in Eq. 3 as DO. Because the original optimization problems over $x$ are equivalent, their DO formulations are equivalent too,

$$\arg\max_\theta \mathbb{E}_{p_\theta(x)}[f(x)] = \arg\max_\theta \left( \mathbb{E}_{p_\theta(x)}[Q_r^{\max}(\tilde{x}_r)] + \sum_{(\mathrm{p}(i),i)\in\mathcal{E}'} \mathbb{E}_{p_\theta(x)}[Q_i^{\max}(\tilde{x}_i, \tilde{x}_{\mathrm{p}(i)})] \right), \quad (4)$$

where we've used linearity of expectations on the right side. However, a generic joint search distribution cannot take advantage of the linear additivity in value functions over the junction tree topology. That is, while the classical traceback equations perform maximization over each meta-variable separately, Eq. 4 uses a single, unfactorized search distribution over *all* variables, $p_\theta(x)$, to optimize each $Q$-function. We address this next, by factorizing the search distribution.

**Factorized search distribution.** Classical message-passing (Eq. 2) independently maximizes each $Q$-function conditional on the choice of $\tilde{x}_{\mathrm{p}(i)}$, instead of explicitly maximizing all variables in $x$ jointly (Eq. 3). This is possible because each $Q_i$ captures all relevant global information needed to choose $\tilde{x}_i$. It stands to reason then that DO can do something similar. Specifically, instead of training a single joint search distribution, we train smaller search distributions over each $\tilde{x}_i$, conditional on $\tilde{x}_{\mathrm{p}(i)}$, to separately maximize each $Q_i^{\max}(\tilde{x}_i, \tilde{x}_{\mathrm{p}(i)})$. That is, we factor the search distribution according to $\mathcal{T}'$, resulting in a DAG, $p_\theta(x) := p_\theta(\tilde{x}_r) \prod_{(\mathrm{p}(i),i)\in\mathcal{E}'} p_\theta(\tilde{x}_i \mid \tilde{x}_{\mathrm{p}(i)})$, for root node $r$, non-root nodes $i$, and parents $\mathrm{p}(i)$, connected by directed edges $\mathcal{E}'$, and with parameters, $\theta$. This factorized search distribution can be plugged into Eq. 4 for an equivalent optimization problem. We refer to each element of this product as one of the *factors* of the search distribution. In our implementation, each factor has completely separate parameters, though we write a shared $\theta$ for conciseness. Notice that each factor of the search distribution interacts with each other factor through the directed edges, $\mathcal{E}'$. That is, the distribution of $\tilde{x}_i$ depends on its parent's factor, and through it, all of its parent's ancestors: $p_\theta(\tilde{x}_i) = p_\theta(\tilde{x}_i \mid \tilde{x}_{\mathrm{p}(i)})p_\theta(\tilde{x}_{\mathrm{p}(i)})$. In turn, each of node $i$'s children's factors depends on $p_\theta(\tilde{x}_i)$. Due to this coupling, we cannot optimize each factor fully independently. But we can still *update* each factor separately from the others in a globally-consistent manner via message-passing. In particular, each factor will only be responsible for directly optimizing its own meta-variable, but will need to coordinate with its neighboring factors by getting information from them about how they are optimizing their meta-variables in a manner analogous to classical message-passing. Our current messages, $Q_i^{\max}$, convey the value of each intermediate objective when all meta-variables are chosen via maximization. For DO, we'll require messages that communicate values when meta-variables are chosen according to the factorized search distribution, $p_\theta(x)$.

**Distributional value functions.** While one certainly could choose to optimize classical value functions using an EDA search distribution (Eq. 4), it doesn't make sense for two main reasons. As we just mentioned, DO aims to train $p_\theta(x)$ such that it maximizes $f(x)$, or equivalently, the sum of $Q$-functions, in expectation. Therefore, the DO objective should consider intermediate objective values for meta-variables chosen according to the current search distribution, not those chosen by explicit maximization, as in $V_i^{\max}(\tilde{x}_{\mathrm{p}(i)})$. Additionally, computing classical value functions requires enumerating all assignments of each $\tilde{x}_i$ for the maximum used in $V_i^{\max}(\tilde{x}_{\mathrm{p}(i)})$. When $\tilde{x}_i$ contains more than a few design variables, this `max` operation quickly becomes intractable. One reason for doing distributional optimization is to avoid enumerating massive design spaces. To address both issues, we define corresponding *distributional* value functions that fulfill both desiderata:

$$V_i^\theta(\tilde{x}_{\mathrm{p}(i)}) := \mathbb{E}_{p_\theta(\tilde{x}_i \mid \tilde{x}_{\mathrm{p}(i)})}[Q_i^\theta(\tilde{x}_i, \tilde{x}_{\mathrm{p}(i)})], \quad \text{where}$$

$$Q_i^\theta(\tilde{x}_i, \tilde{x}_{\mathrm{p}(i)}) := f_{\mathrm{p}(i),i}(\tilde{x}_{\mathrm{p}(i)}, \tilde{x}_i) + Q_i^\theta(\tilde{x}_i) \text{ and } Q_i^\theta(\tilde{x}_i) := f_i(\tilde{x}_i) + \sum_{c\in\mathrm{children}(i)} V_c^\theta(\tilde{x}_i).$$

Compared to Eq. 1, the distributional $V$-functions compute the value in expectation under $p_\theta$ instead of a max. Because they use an expectation instead of a max operation, these value functions can be approximated tractably and without bias by drawing Monte-Carlo samples from the search distribution. Moreover, each distributional value function lower bounds each corresponding classical value function because the expectation of a function cannot exceed its maximum (details in Sec. A.4). As a result, the sum of classical value functions in the objective in Eq. 4 is bounded below by the sum

of distributional value functions, which we'll optimize instead:

$$\mathbb{E}_{p_\theta(x)}[Q_r^{\max}(\tilde{x}_r)] + \sum\nolimits_{(\mathrm{p}(i),i)\in\mathcal{E}'} \mathbb{E}_{p_\theta(x)}[Q_i^{\max}(\tilde{x}_i, \tilde{x}_{\mathrm{p}(i)})] \geq$$
$$\mathbb{E}_{p_\theta(x)}[Q_r^\theta(\tilde{x}_r)] + \sum\nolimits_{(\mathrm{p}(i),i)\in\mathcal{E}'} \mathbb{E}_{p_\theta(x)}[Q_i^\theta(\tilde{x}_i, \tilde{x}_{\mathrm{p}(i)})]. \quad (5)$$

**Distributional optimization with value functions.** All that remains is to derive an update rule that treats each search distribution factor separately. Since each expectand in Eq. 5 doesn't depend on descendant meta-variables, we can replace $p_\theta(x)$ with each meta-variable's marginal distribution:

$$\arg\max_\theta \mathbb{E}_{p_\theta(\tilde{x}_r)}[Q_r^\theta(\tilde{x}_r)] + \sum\nolimits_{(\mathrm{p}(i),i)\in\mathcal{E}'} \mathbb{E}_{p_\theta(\tilde{x}_i|\tilde{x}_{\mathrm{p}(i)})p_\theta(\tilde{x}_{\mathrm{p}(i)})}[Q_i^\theta(\tilde{x}_i, \tilde{x}_{\mathrm{p}(i)})].$$

We then follow the EDA derivation to arrive at an update rule (details in Sec. A.4) in which each term is optimized by only a single factor; dependence on $p_\theta(\tilde{x}_{\mathrm{p}(i)})$ is approximated by sampling. We write DADO's update rule (sharing a single set of samples; see Fig. 1b) as a sum of weighted likelihoods for each search distribution factor,

$$\theta^{n+1} = \arg\max_\theta \sum_{x \sim p_{\theta^n}(x)} \left( Q_r^{\theta^n}(\tilde{x}_r) \log p_\theta(\tilde{x}_r) + \sum_{(\mathrm{p}(i),i)\in\mathcal{E}'} Q_i^{\theta^n}(\tilde{x}_i, \tilde{x}_{\mathrm{p}(i)}) \log p_\theta(\tilde{x}_i \mid \tilde{x}_{\mathrm{p}(i)}) \right),$$

which is equivalent to separate updates because the factors don't share parameters, as desired:

$$\theta_r^{n+1} = \arg\max_{\theta_r} \sum_{x \sim p_{\theta^n}(x)} Q_r^{\theta^n}(\tilde{x}_r) \log p_{\theta_r}(\tilde{x}_r), \quad \text{and}$$
$$\theta_i^{n+1} = \arg\max_{\theta_i} \sum_{x \sim p_{\theta^n}(x)} Q_i^{\theta^n}(\tilde{x}_i, \tilde{x}_{\mathrm{p}(i)}) \log p_{\theta_i}(\tilde{x}_i \mid \tilde{x}_{\mathrm{p}(i)}), \quad \forall\, i \in \mathcal{N} \setminus \{r\}.$$

Each factor is weighted by its corresponding value function, enabling it to coordinate with all its descendant factors despite their being updated separately. The whole DO is tied together at the top by the root factor. We emphasize that DADO's update is more statistically efficient than a naive EDA's because DADO gets to use all $K$ samples for weighted MLE on each lower-dimensional factor distribution (*i.e.*, ratio of number of samples to number of dimensions is larger). Our resulting algorithm (Alg. 1) fits into the three EDA steps: (1) designs are sampled from $p_{\theta^n}(x)$, (2) weights, here each $Q$-function instead of $f(x)$, are computed, and (3), each search distribution factor receives its own independent weighted maximum likelihood update based on these weights (Fig. 1b). These are repeated until convergence, or for some fixed number of iterations. The factor updates are coupled only through the $Q$-functions, the messages across edges. $\{Q_i^{\theta^n}\}_{i\in\mathcal{N}}$ are only valid while $\theta$ is close to $\theta^n$, meaning one must balance how many gradient steps are taken before drawing new samples. If too many gradient steps are taken without resampling, a factor may be changing its parameters to collaborate with another factor which has already changed its behavior. It's common for EDAs to include an additional hyperparameter $W(\cdot)$, a monotonic shaping function applied to the weights, to alter optimization dynamics (Brookes et al., 2020), included in Alg. 1. Choosing $W$ to be the identity recovers our derivation. Notice that in place of the classical message-passing traceback equations, DADO simply performs sequential conditional sampling from its search distribution. Interestingly, a very similar algorithm can be derived without using a lower bound on the value functions (Eq. 5) if one adds an entropy-maximizing term to the initial DO objective (Sec. A.5).

## 3 RELATED WORK

The closest work to ours in the EDA community is that of the Factorized Distribution Algorithm (FDA), wherein a decomposition is leveraged to factorize the conditional probability table (CPT) that parameterizes the search distribution (Mühlenbein & Mahnig, 1999). However, FDA cannot leverage message passing to give different weights to different variables so as to more efficiently update the search model. In related work, Pelikan (2005) replaced the CPTs with Bayesian networks, but still had no means to coordinate among variables through message passing or the like.

In a complementary line of work on policy optimization in reinforcement learning (analogous to learning the search distribution in EDAs), coordination between variables is enabled by message passing on a factorized search distribution/policy, but this line of work is only suitable for graph topologies that are Markov chains—that is, chain graphs. These graphs arise from reward functions,

---

**Algorithm 1** DECOMPOSITION-AWARE DISTRIBUTIONAL OPTIMIZATION (DADO)

---

1: **Given:** junction tree $\mathcal{T} = (\mathcal{N}, \mathcal{E})$, component functions $\{f_{i,j}(\tilde{x}_i, \tilde{x}_j)\}_{(i,j)\in\mathcal{E}}$ and $\{f_i(\tilde{x}_i)\}_{i\in\mathcal{N}}$

2: Root $\mathcal{T}$ such that tree height is minimized, yielding directed junction tree, $\mathcal{T}' = (\mathcal{N}, \mathcal{E}')$

3: Factorize search distribution according to $\mathcal{T}'$ as $p_\theta(x) \coloneqq p_\theta(\tilde{x}_r) \prod_{(\mathrm{p}(i),i)\in\mathcal{E}'} p_\theta(\tilde{x}_i \mid \tilde{x}_{\mathrm{p}(i)})$; initialize $\theta$

4: Sort nodes topologically: order $O_\mathcal{N}$ such that $O_\mathcal{N}^{(0)}$ is a leaf index and $O_\mathcal{N}^{(L-1)}$ is root index

5: **for** iteration $n = 1, 2, \ldots, N$ **do**

6: $\quad \{\tilde{x}^k\}_{k=1}^K \sim p_\theta(x)$ $\qquad\qquad\qquad$ ▷ Sample from the factorized search distribution

7: $\quad$ **for** $i \in O_\mathcal{N}$ **do** $\qquad\qquad$ ▷ Estimate distributional value functions from samples

8: $\quad\quad$ **for** $k = 1, 2, \ldots K$ **do**

9: $\quad\quad\quad Q_i(\tilde{x}_i^k) \leftarrow f_i(\tilde{x}_i^k) + \sum_{c\in\mathrm{children}(i)} \sum_{k=1}^K [Q_c(\tilde{x}_c^k, \tilde{x}_i^k)]$

10: $\quad\quad\quad$ **if** $i$'s parent, $\mathrm{p}(i)$, exists:

11: $\quad\quad\quad\quad Q_i(\tilde{x}_i^k, \tilde{x}_{\mathrm{p}(i)}^k) \leftarrow f_{\mathrm{p}(i),i}(\tilde{x}_{\mathrm{p}(i)}^k, \tilde{x}_i^k) + Q_i(\tilde{x}_i^k)$

12: $\quad\quad$ **end for**

13: $\quad$ **end for** $\qquad\qquad$ ▷ Update each search distribution factor using value functions as weights

14: $\quad$ **for all** $i \in \mathcal{N}$, in parallel **do**

15: $\quad\quad$ **if** $i = r$:

16: $\quad\quad\quad \theta_r \leftarrow \arg\max_{\theta_r} \sum_{k=1}^K W(Q_r(\tilde{x}_r^k)) \log p_{\theta_r}(\tilde{x}_r^k)$

17: $\quad\quad$ **else**

18: $\quad\quad\quad \theta_i \leftarrow \arg\max_{\theta_i} \sum_{k=1}^K W(Q_i(\tilde{x}_i^k, \tilde{x}_{\mathrm{p}(i)}^k)) \log p_{\theta_i}(\tilde{x}_i^k \mid \tilde{x}_{\mathrm{p}(i)}^k)$

19: $\quad$ **end for**

20: **end for**

---

$f(x)$, that decompose in time according to a first-order Markov assumption (Peters & Schaal (2007), Peng et al. (2019), Nair et al. (2020)). Such approaches cannot be used on arbitrary junction trees.

Adjacent to our problem of interest, because they focus solely on real-valued design spaces, Grudzien et al. (2024) introduces a new way to discover a functional decomposition from data, and then demonstrate that doing so helps improve out-of-distribution generalization. Separately, the Bayesian optimization (BO) community has developed methods for dynamically inferring a function decomposition from data during active learning. Although they make use of message passing for coordination, they do not employ distributional optimization, nor can these methods be readily generalized to do so (Kandasamy et al. (2015), Han et al. (2021), Hoang et al. (2018), Rolland et al. (2018), Bardou et al. (2024)). This community has shown that using approximate or even random decompositions can be helpful (Ziomek & Ammar, 2023). DADO could, in principle, be used within the BO inner loop, although such an investigation is beyond the scope of the present work.

## 4 EXPERIMENTAL RESULTS

We perform two sets of experiments with increasing resemblance to real-world scientific design. In each setting we compare a standard EDA, that is unaware of the function decomposition, to DADO, which is aware of it. In the first setting, we create synthetic functions $f(x)$, while in the second setting, we focus on functions derived from real protein data. In all experiments we designed sequences of fixed length $L$, where each position is one of $D = 20$ amino acids. We used $N = 100$ EDA training iterations and $G = 1$ gradient steps for each weighted maximum likelihood update. We use MLP-based search distributions (details in Sec. A.8.1) for DADO and all baselines. We also compare to FDA (Mühlenbein & Mahnig, 1999) and PPO (Schulman et al., 2017) (details in Sec. A.7). For the shaping function we choose $W(s) = \exp\frac{s}{\beta}$ and tune the temperature, $\beta$. To pick $\beta$ and the learning rate, $\eta$, we performed a hyperparameter sweep separately for each method and each $f(x)$, taking $(\eta^*, \beta^*)$ with the largest sample mean at the last iteration. For synthetic experiments, we swept over 100 combinations between $\eta \in [10^{-5}, 5 \times 10{-2}]$ and $\beta \in [0.1, 8]$. For protein experiments, which take longer, we swept only 54 pairs between $\eta \in [5 \times 10^{-5}, 5 \times 10{-2}]$ and $\beta \in [0.01, 5]$. If either method's $\eta^*$ or $\beta^*$ was on the boundary of the

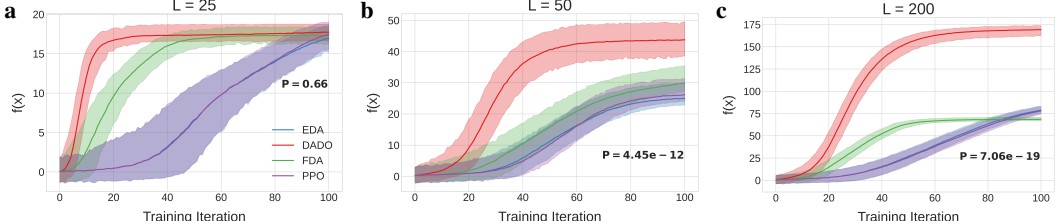

**Figure 2: Comparison of a naive EDA to DADO on synthetic problems.** We created three random functions, $f(x)$, each with a randomly chosen junction tree decomposition with maximum node size of one, and randomly chosen parameters. Each experiment used alphabet size $D = 20$ and sequence length, **a**, $L = 25$, **b**, $L = 50$, and **c**, $L = 200$. Each of the two methods drew $K = 100$ samples per iteration. For each iteration, we show the mean (solid line) and 95% confidence interval (shaded envelope) of the 100 samples evaluated on $f(x)$, averaged across results from 20 random seeds. P-values are from a two-sided paired t-test that the mean at the final iteration is different between DADO and the most competitive baseline, over the 20 seeds.

sweep, we broadened it. $(\eta^*, \beta^*)$ were then used for all replicate runs over random seeds dictating the initial search distribution and sampling. Although not essential to our method as described in Alg. 1, in our implementation we reduce the variance of the search distribution update by using mean-shifted $Q$-functions, $Q_i(\tilde{x}_i, \tilde{x}_p) - \mathbb{E}_{\{\tilde{x}_i^k\}}[Q_i(\tilde{x}_i^k, \tilde{x}_p)]$, which are unbiased (Williams, 1992). Statistical significance of differences in performance is computed by comparing the sample mean $f(x)$ at the final iteration, using paired two-sided t-tests over random seeds.

## COMPARISON ON FULLY SYNTHETIC FUNCTIONS

Here we used junction trees with meta-variable nodes containing only one original variable, *i.e.*, an exact tree decomposition. We did so because this allowed us to better control the difficulty of the synthetic functions in that we could specify each component function as a dense lookup table of size $D^2$ and thereby ensure that there would be non-smoothness all throughout the design space. In contrast, larger nodes would have increased lookup table sizes exponentially with the number of variables in each meta-variable, requiring a more compact representation. Sparse lookup tables are one such option but generally result in needle-in-a-haystack-like functions. Neural networks are another, but random initialization tends to yield overly smooth functions that are unrealistically easy to optimize. Nor is it obvious how to produce neural landscapes that are realistically difficult across the entire design space via training. We simulated one $f(x)$ for each of three sequence lengths, $L = \{25, 50, 200\}$, by randomly sampling a junction tree and component functions (details in Sec. A.8.2). Search model weights were initialized from $\mathcal{N}(0, 0.0004)$ and biases were set to 0.

All methods drew $K = 100$ samples from their search distribution at each iteration. For all three sequence lengths, DADO outperforms all three baselines (Fig. 2). When $L = 25$, the baselines catch up to DADO by the end of 100 iterations, but for larger $L$, corresponding to larger design spaces, the baseline methods converge to substantially lower values of $f(x)$ than DADO. Results from experiments in which we increase $D$ further support DADO's superiority (Fig. A3).

## COMPARISON ON PROTEIN PROPERTY PREDICTORS

Here we anchor our experiments on $f(x)$ fit to real protein datasets. We use four protein property datasets, AAV, Amyloid, Gcn4, and TDP-43 (details in App. A.6). For each dataset, we used AlphaFold3-predicted structures (Abramson et al., 2024) on the wild-type sequence to obtain a 3D structure, from which we constructed a contact graph by thresholding the distance between pairs of residues with threshold $t$. Following Brookes et al. (2022); Romero et al. (2013); Voigt et al. (2002), we use a threshold of $t = 4.5$Å. We interpret this contact map as a graph adjacency matrix, from which we algorithmically construct a junction tree (Lauritzen & Spiegelhalter, 1988). This defines the topology needed for DADO (Fig. A1), and then we fit the component functions on the protein assay-labeled data (details in Sec. A.8.3). We initialized the search distribution by training on the 1,000 sequences with lowest $f(x)$, for 1,000 iterations, with a learning rate of $2 \times 10^{-3}$.

All methods drew $K = 1000$ samples from their search distribution at each iteration. For all four proteins, DADO converges to designs with higher $f(x)$ than the baselines both visually and by a statistical test (Fig. 3). For three of the four (Amyloid, Gcn4, and TDP-43), the seed-averaged 95% confidence intervals of DADO and the best baseline at the final iteration do not overlap. We plotted

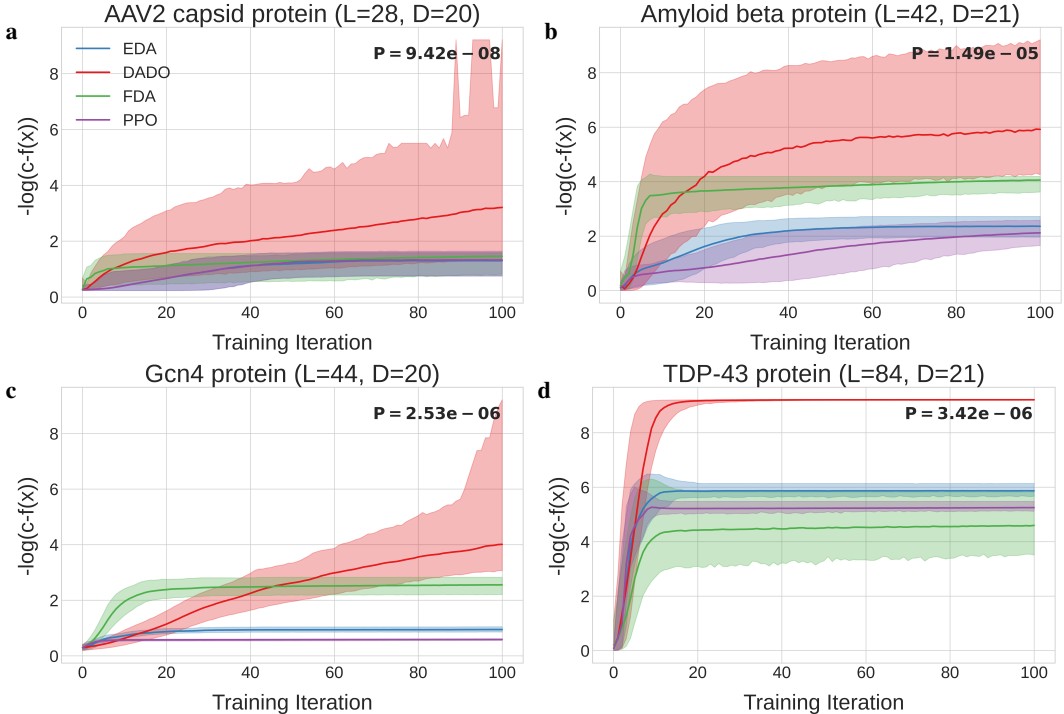

**Figure 3: Comparison of a naive EDA to DADO on protein property predictive models.** For each of four proteins of varying length, **a,** AAV, **b,** Amyloid, **c,** Gcn4, and **d,** TDP-43, we fit a neural network property function, $f(x)$, adhering to a junction tree decomposition derived from the protein's 3D structure, and then used a naive EDA and DADO to optimize them. Each approach drew $K = 1000$ samples per EDA iteration. For each iteration, we show the mean (solid line) and 95% confidence interval (shaded envelope) of the 1000 samples evaluated on $-\log(c - f(x))$, averaged across results from 20 random seeds. We plot this quantity to make clear the differences between methods when $f(x)$ is large; $c$ is the largest $f(x)$ on a given plot, plus a small constant for numerical stability. P-values are from a two-sided paired t-test that the mean at the final iteration is different between DADO and the most competitive baseline, over the 20 seeds.

$-\log(c - f(x))$ on the y-axis to make this difference clear visually; plots with $f(x)$ on the y-axis are included in the appendix (Fig. A5). Finally, in Sec. A.9.5, we observe that our choice of AlphaFold3-based topology is robust to mild modifications as far as holdout performance is concerned.

## 5 DISCUSSION

We have proposed a new method for distributional optimization that can leverage arbitrary decomposability of the function being optimized. We have shown that it works as expected on synthetic problems–namely, better than a naive EDA that is not aware of the decomposition. We have also demonstrated the potential for practical utility on the problem protein design. Importantly, it is not necessary that a problem strictly adhere to the specified decomposition in order to be useful. Specifically, we showed that using the heuristic of thresholding AlphaFold3 estimated contacts, we can obtain a range of decomposed functions, including some that maintain predictive accuracy while providing a level of decomposability helpful for DADO. Obtaining similarly useful decomposability in other domains will require further investigation; however, as the real world typically is structured, we expect many areas will be amenable to doing so. Estimating decomposability from labeled data is an active area of research (Poelwijk et al. (2016); Grudzien et al. (2024); Park et al. (2024)).

It will be insightful to conduct future work interrogating the use of DADO with generalizations of the standard EDA, such as conditioning an unconditional generative model with an inaccurate property predictor for AI-guided design, which should enable us to design while accounting for the fact that $f(x)$ is not truly known everywhere in the space due to finite data. This particular problem may require distilling an unconditional model into one that respects the decomposition, although alternative strategies could prove useful. Finally, use of DADO within Bayesian Optimization for optimization of the acquisition function is an exciting direction.

## 6 REPRODUCIBILITY STATEMENT

We specify exhaustively our experimental setup, hyperparameters, model architectures, and algorithm details in Sec. 4. Datasets used and their sources are detailed in Sec. A.6. Code for our method and experiments is available publicly on GitHub: `https://github.com/james-bowden/DADO`.

## ACKNOWLEDGMENTS

We thank Kuba Grudzien for helpful discussions regarding function decomposition, Hanlun Jiang and Peter Ma for discussions regarding scientific applications, Toby Kreiman and Bear Xiong for feedback on the manuscript, and Aileen Zhang for feedback on the figures.

JCB was supported in part by an NSF GRFP fellowship and a UC Berkeley Chancellor's fellowship. This material is based upon work supported by the National Science Foundation Graduate Research Fellowship Program under Grant No. DGE 2146752. Any opinions, findings, and conclusions or recommendations expressed in this material are those of the author(s) and do not necessarily reflect the views of the National Science Foundation. JCB and JL were supported in part by the Office of Naval Research (ONR) under grant N00014-23-1-2587, and the U.S. Department of Energy, Office of Science, Office of Biological and Environmental Research, Lawrence Livermore National Laboratory BioSecure SFA within the Secure Biosystems Design program (SCW1710).

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

## A APPENDIX

### A.1 DIFFERENT WAYS OF WRITING A FUNCTION DECOMPOSITION

In the introduction (Sec. 1), we define a decomposable function as one that can be written like $f(x) = C_1(\hat{x}_1) + C_2(\hat{x}_2), \ldots, C_\kappa(\hat{x}_\kappa)$, with meta-variables $\hat{x}_i$ that are generic sets of original variables $x_j$. This formulation is most helpful for building intuition about fully-decomposable functions, in which there are no variables $x_j$ that appear in multiple meta-variables (*i.e.*, meta-variables don't overlap). It also encompasses functions for which there are overlapping meta-variables in the decomposition too.

$f(x) = C_1(\hat{x}_1) + C_2(\hat{x}_2), \ldots, C_\kappa(\hat{x}_\kappa)$ also provides the helpful intuition that someone looking to use DADO need only specify a decomposition at this level of detail—sets of design variables that directly interact—rather than as a graph. In the protein binding example given in Sec. 1, a user might specify three component functions: one over sequence positions in the binding interface, $\hat{x}_0$, one over sequence positions in the scaffold, $\hat{x}_1$, and one over a subset of positions tying together the binding interface and the scaffold, $\hat{x}_2$. Specifying a decomposition at this level may be easier than in its graph form. In the event that one has an exact decomposition given, this information can be represented either this way, or in terms of the graph introduced below. Situations in which an exact decomposition may be given include hardware design (*e.g.*, due to manufacturing constraints, a topology of wires in a circuit is pre-specified) and bi-level optimizations (*e.g.*, to design the best telescope, a scientist optimizes both the arrangement of lenses and their physical properties; an outer loop optimizes topologies and an inner loop using DADO evaluates a single topology by optimizing each lens' parameters).

However, $f(x) = C_1(\hat{x}_1) + C_2(\hat{x}_2), \ldots, C_\kappa(\hat{x}_\kappa)$ as written has no interpretation in terms of a graph, which is necessary for message-passing and for factorizing the search distribution (Sec. 2.1). To bridge this gap, we introduce a graph in which nodes are design variables and edges specify which design variables directly interact to influence $f$. Users of DADO may also specify the decomposition at this level; we expect this to be helpful when domain knowledge is available in the *negative* form, *i.e.*, when the user can begin with the fully-connected graph and remove edges between design variables they think don't directly interact. This graph can then be automatically transformed into a junction tree (Lauritzen & Spiegelhalter, 1988), which is what our algorithm actually takes as input. That is, DADO takes an undirected junction tree, $\mathcal{T} := (\mathcal{N}, \mathcal{E})$, with nodes, $\mathcal{N}$, and edges, $\mathcal{E}$. Each node is a set of design variable indices, such that indexing $x$ with this set yields a new meta-variable, $\tilde{x}_i$, correspondingly exactly to a node in the junction tree, whereas the old meta-variables $\hat{x}_i$ had no such interpretation. $f(x)$ must be decomposed instead according to $\mathcal{T}$.

We can rewrite the generic function decomposition, $f(x) = C_1(\hat{x}_1) + C_2(\hat{x}_2), \ldots, C_\kappa(\hat{x}_\kappa)$, equivalently in terms of $\mathcal{N}$ and $\mathcal{E}$: $f(x) = \sum_{i \in \mathcal{N}} f_i(\tilde{x}_i) + \sum_{(i,j) \in \mathcal{E}} f_{i,j}(\tilde{x}_i, \tilde{x}_j)$ (Fig. 1a). We've replaced component functions $C_i(\hat{x}_i)$ on generic sets of design variable indices with either an equivalent "node component function", $f_i(\tilde{x}_i)$, or "edge component function", $f_{i,j}(\tilde{x}_i, \tilde{x}_j)$. Correspondingly, $\hat{x}_i$ in the original formulation either corresponds exactly to some node in the junction tree, such that $\hat{x}_i = \tilde{x}_j$, or it corresponds to some edge in the junction tree, such that $\hat{x}_i = \tilde{x}_{j \cup k}$. The fully-decomposed, no-overlap EDA described in Sec. 2 corresponds to an edgeless junction tree— each component function is simply a node component function, $C_i(\hat{x}_i) = f_i(\tilde{x}_i)$, and there are no edge component functions—which is why a fully-factorized search distribution suffices. But if, for example, $\hat{x}_0 = \{x_0, x_1\}$ and $\hat{x}_1 = \{x_1, x_2\}$ overlap, then they must instead correspond to edge component functions. For a simplified case where the junction tree nodes have cardinality 1 (*e.g.*, $\tilde{x}_i = x_i$), we might have $C_0(\hat{x}_0) = f_{0,1}(\tilde{x}_0, \tilde{x}_1)$ and $C_1(\hat{x}_1) = f_{1,2}(\tilde{x}_1, \tilde{x}_2)$. Fig. 1a only depicts edge component functions for clarity and simplicity; notice also that node component functions can technically be subsumed into appropriate edge component functions without loss of generality. The presence of edge component functions requires the coupling of the search distribution's factors in a manner compatible with $\mathcal{E}$; a fully-factorized search distribution with independent distributions for each meta-variable will no longer suffice.

### A.1.1 CONNECTION TO EPISTASIS

Function decomposition corresponds intimately to notions of the epistatic landscape for a protein property function (*e.g.*, Poelwijk et al. (2016), Wu (1982), Otwinowski & Nemenman (2013), Lipsh-

Sokolik & Fleishman (2024)). An "epistatic expansion" of $f$,

$$f(x) = \beta + \sum_{i=0}^{L} \beta_i[x_i] + \sum_{i=0}^{L}\sum_{j>i}^{L} \beta_{i,j}[x_i, x_j] + ... + \beta_{0,1,...L-1}[x_0, x_1, ..., x_{L-1}], \qquad (6)$$

is a decomposition of $f$ into a bias term, first-order terms, second-order terms, and so forth up to $L$-order epistasis. For the first-order terms, each $\beta_i$ is a vector which is indexed by the particular value of $x_i$. For higher-order terms, $\beta$ is a $d$-dimensional tensor. One can obtain a decomposition graph from an epistatic expansion as follows: for each nonzero term in the expansion, add an edge between each design variable in the term. A function in which only a few, lower-order terms were nonzero would be relatively easy to optimize.

An expansion of this sort can be thought of as a spectral decomposition of $f$ into components of lower and higher frequencies, and can be computed for small landscapes using a discrete Fourier transform. A function with primarily lower-frequency components would be smooth and easy to optimize whereas a function with higher-frequency components would be rugged and difficult to find the global optimum of. One can imagine a worst-case scenario, that of $L$-order epistasis, of which a needle-in-a-haystack function is an example. In general, full epistatic landscapes are intractable to compute for design spaces of practical sizes, but estimation of a subset of the terms is an active area of research (*e.g.*, Otwinowski & Nemenman (2013); Park et al. (2024)) due to beliefs that natural proteins tend to exhibit sparse and mostly lower-order epistasis.

In comparison to the function decompositions used by DADO, a full epistatic landscape specifies both the graph topology and the component functions, everywhere. Completely specifying an order-$d$ interaction if each variable has $D$ states requires choosing $D^d$ values. The high-dimensional synthetic functions used in our experiments are fully specified everywhere because they're defined with primarily lower-order epistasis, which can be specified with far fewer values. In contrast, our protein property predictors can have high-order component functions represented by neural networks fit from limited data, meaning they're underspecified beyond the training data. In some sense, fitting neural network component functions allows us to specify higher-order epistasis in a locale without paying the price for defining it over the full design space. Real-world design procedures typically only have knowledge of $f$ in a small locale, which is why locally-valid decompositions suffice.

## A.2 DECOMPOSITION-UNAWARE EDA DERIVATION

Given an objective, $f(x)$, to maximize, a decomposition-unaware EDA (or "naive" EDA) transforms the original problem into a distributional optimization problem as follows (Brookes et al., 2020):

$$\max_{x} f(x) = \max_{\theta} \mathbb{E}_{p_\theta(x)}[f(x)]. \qquad (7)$$

For the equivalence to hold, the search distribution, $p_\theta(x)$, must be capable of representing a point mass on $x^*$. Intuitively, one can think of an EDA as having a search distribution, $p_\theta(x)$, that acts as a spotlight on the design space, which is iteratively moved toward areas of the space with high $f(x)$ in expectation. Modern EDAs parameterize the search distribution with neural network generative models, such as a Variational Autoencoder (Kingma & Welling (2013); *e.g.*, Brookes et al. (2019)).

Procedurally, after having initialized the search distribution, $p_{\theta^0}(x)$, this naive EDA then iterates through these three steps either $N$ times or until some convergence criteria is met (Larrañaga & Lozano, 2001; Brookes et al., 2020):

1. Draw $K$ samples from the current search distribution, $\{x^k\}_{k=1}^{K} \sim p_{\theta^n}(x)$

2. Score each sample with the objective function to get its weight, $w^k = f(x^k)$.

3. Update the search distribution parameters with weighted maximum likelihood estimation (MLE), using the weighted samples, $\theta^{n+1} = \arg\max_\theta \mathbb{E}_{\{x^k\}}[w^k \log p_\theta(x^k)]$. In older EDAs, this step was often instead a *truncated* maximum likelihood estimation, in which $p_\theta(x)$ modeled only samples with the largest weights.

Often, a predefined monotonic "shaping" function, $W(\cdot)$, is additionally applied to the weights to control optimization dynamics. Its monotonicity guarantees that

$\arg\max_x f(x) = \arg\max_x W(f(x))$. We've written the EDA above and its derivation below without $W$, but $f(x)$ can be equivalently replaced with $W(f(x))$ everywhere.

For clarity, we sketch out one way of deriving the EDA update (step 3) from Eq. 7. At each EDA iteration, we want to improve the current search distribution with an update, $p_{\theta^{n+1}}(x) \leftarrow p_n^*(x) \propto f(x) \cdot p_{\theta^n}(x)$. To do this, we maximize

$$-D_{\mathrm{KL}}(p_n^*(x) \,\|\, p_\theta(x)) = \frac{1}{Z}\mathbb{E}_{p_{\theta^n}(x)}[f(x)\log p_\theta(x)] + H(p_n^*(x)), \tag{8}$$

where the entropy term can be dropped because it has no dependence on $\theta$ and division by $Z$ can be dropped without changing the objective's maximizer. The sample approximation of the remaining terms is exactly step 3. Assuming an exact update (*i.e.*, infinite samples such that the expectation is evaluated exactly, $p_\theta(x)$ has sufficient capacity to represent $p_n^*(x)$, and the KL divergence reaches 0), the resulting search distribution at iteration $n$ is $p_n^*(x) \propto f(x)^n p_{\theta^0}(x)$. As $n \to \infty$, this distribution concentrates all of its mass at the global maxima of $f$ and as a result, $f(x^*) = \mathbb{E}_{p_\theta(x)}[f(x)]$. The derivation in Brookes & Listgarten (2018) is perhaps closest to this one, though their search distribution improvement operator is motivated by Bayes' rule. It can be generalized to the derivation we give by simply writing $p_n^*(x) \propto C(x) \cdot p_{\theta^n}(x)$ for some $C$. Bayes' rule yields a special case of this improvement operator, namely one where the multiplier is $C(x) = \frac{p(S|x)}{\sum_x p_{\theta^n}(x)p(S|x)}$. We choose not to interpret $f(x)$ as a normalized distribution in our derivation, though one can definitely do so.

In practice, the weighted MLE problem in step 3 is typically solved with gradient descent; we will choose to use only a fixed number of gradient steps, which can be theoretically justified through the equivalence of EDAs to Expectation-Maximization (Brookes et al., 2020).

### A.3 ALGORITHMIC SKETCH: FROM EDA TO DADO

In this section, we describe how infusing a naive EDA (Sec. A.2) with awareness of a known decomposition of $f(x)$ stands to make the EDA more statistically efficient.

Intuitively, one can think of an EDA as having a search distribution, $p_\theta(x)$, that acts as a spotlight on the design space, which is iteratively moved toward areas of the space with high $f(x)$ in expectation. In a decomposition-unaware EDA, after having initialized the search distribution, $p_{\theta^0}(x)$, with initial parameters, $\theta^0$, the EDA proceeds by iterating through these three steps for $n = 1 \ldots N$ or until some convergence criteria has been met (Sec. A.2):

1. Draw $K$ samples from the current search distribution, $\{x^k\}_{k=1}^K \sim p_{\theta^n}(x)$

2. Score each sample with the objective function to get its weight, $w^k = f(x^k)$.

3. Update the search distribution parameters with weighted maximum likelihood estimation (MLE), using the weighted samples, $\theta^{n+1} = \arg\max_\theta \mathbb{E}_{\{x^k\}}[w^k \log p_\theta(x^k)]$.

Now that a standard EDA is fresh in our minds, we can consider what it might mean to infuse it with knowledge of how $f(x)$ is decomposed, and how this might prove useful. Before discussing the general case, here we first build intuition by considering a simpler example, one that is "fully"-decomposable rather than only soft-decomposable. That is, consider $f(x) = C_1(\hat{x}_1) + C_2(\hat{x}_2)$ where the two meta-variables $\hat{x}_1$ and $\hat{x}_2$ do not share any original variables, $x_i$, corresponding to a junction tree without any edges, only independent nodes. In this setting, we can break the EDA down into two independent EDAs, each with reduced dimensionality compared to the original EDA. Specifically, we can employ a fully-factorized search distribution ("fully" again denoting that no variable appears in more than one factor), $p_\theta(x) = p_{\theta_1}(\hat{x}_1)p_{\theta_2}(\hat{x}_2)$. We modify the three steps of the EDA as follows:

1. Instead of drawing samples from one search distribution over all design variables, we sample each meta-variable from its own search distribution separately, $\hat{x}_1^k \sim p_{\theta_1^n}(\hat{x}_1)$ and $\hat{x}_2^k \sim p_{\theta_2^n}(\hat{x}_2)$.

2. Instead of scoring $x^k$ with $f(x)$, we score each meta-variable separately with its component function, for $i \in \{1, 2\}$, to obtain separate weights.

3. Instead of one update to a joint search distribution, we update each factor independently of the others with weighted MLE on samples from the relevant search distribution, $\theta_i^{n+1} = \arg\max_{\theta_i} \mathbb{E}_{\{\hat{x}_i^k\}}[w_i^k \log p_{\theta_i}(\hat{x}_i^k)]$, for $i \in \{1, 2\}$.

What has this bought us? We have split one optimization problem over a combinatorial space into two independent optimization problems, each over a smaller combinatorial space. For example, for binary design variables, where each of the two meta-variables comprises say 15 variables, we have transformed one joint optimization over a space of size $2^{30} \approx 10^9$ to two independent optimization problems each of size $2^{15} \approx 10^5$, thereby reducing the effective search space size by four orders of magnitude. For larger alphabets and sequence lengths, these differences will be larger still. Importantly, the weighted MLE update with a fixed number of samples, $K$, is more statistically efficient for each lower-dimensional search distribution factor than it is for a decomposition-unaware EDA using a single joint search distribution over all design variables.

In the general case, that is, beyond full decomposability, the same variable, $x_i$, can participate in multiple component functions $C_j(\hat{x}_j)$. In other words, the meta-variables $\hat{x}_j$ may overlap, corresponding to there being edges in our junction tree. Because of this "component coupling", the EDA can no longer independently optimize each component function and will require coordination between meta-variables. That is, we can no longer simply divide and conquer as above; rather, we must divide, and then collaborate to conquer. Collaboration among meta-variables will occur by way of a new form of message-passing that allows us to perform distributional optimization while leveraging the decomposability reflected by the junction tree.

Earlier we described decomposability as $f(x) = C_1(\hat{x}_1) + C_2(\hat{x}_2), \ldots, C_\kappa(\hat{x}_\kappa)$ to help provide some intuition. However, to develop DADO it will be more convenient to equivalently describe the decomposability with a junction tree defined by nodes that contain multiple design variables and undirected edges that connect the nodes. We define this junction tree decomposition in Sec. 2.1 and explain its connection to the above decomposability in Sec. A.1. It suffices to say here that design variables $x_i$ that occurred uniquely in one meta-variable $\hat{x}_j$ now sit in a single node, $\tilde{x}_k$; variables that previously occurred in multiple meta-variables now appear in multiple adjacent nodes. With the idea of a junction tree in hand, we are now set to complete our intuitive overview of DADO, which will use a factorized search distribution matching the topology of the junction tree. Namely, we will convert the undirected junction tree into a directed, acyclic graph (DAG) by rooting it and then directing the edges outward from the root. This operation will yield a search distribution that can be written as $p_\theta(x) = p_\theta(\tilde{x}_r) \sum_i p_\theta(\tilde{x}_i \mid \tilde{x}_{\mathrm{p}(i)})$, for root node $\tilde{x}_r$ and non-root nodes, $\tilde{x}_i$, with parents, $\tilde{x}_{\mathrm{p}(i)}$. Note that each non-root node has exactly one parent in a DAG. Each search distribution factor is a conditional probability distribution that will be parameterized by a neural network such as an autoregressive model.

Consequently, sampling from such a factorized search distribution is both computationally efficient and easy, so step 1 of the EDA remains straightforward. However, steps 2 and 3 of the EDA are not so easily generalized. For these steps to make use of the decomposability, we will need to leverage a special kind of message-passing to coordinate updates to different factors of the search distribution so as to achieve a globally consistent update for the entire search distribution. At a high level, our decomposition-aware EDA will proceed as follows for the $n + 1^{\mathrm{th}}$ iteration, compared to the fully-decomposable EDA presented previously:

1. Instead of sampling each meta-variable from an independent search distribution factor, we sample meta-variables sequentially from conditional search distribution factors, starting with the root node, $\tilde{x}_r^k \sim p_{\theta^n}(\tilde{x}_r)$, and conditionally sampling down to the leaves, $\tilde{x}_i^k \sim p_{\theta^n}(\tilde{x}_i \mid \tilde{x}_{\mathrm{p}(i)}^k)$.

2. Instead of scoring each meta-variable $\tilde{x}_i^k$ separately with its component function, $C_i(\tilde{x}_i^k)$, we use message-passing on the junction tree to obtain scores accounting for each meta-variable's coupling to its parents and children, $w_r^k = Q_r(\tilde{x}_r^k)$ for the root and $w_i^k = Q_i(\tilde{x}_i^k, \tilde{x}_{\mathrm{p}(i)}^k)$ for non-root nodes.

3. We perform a coordinated update of the search distribution via per-factor weighted MLE on the relevant samples, $\theta_r^{n+1} = \arg\max_{\theta_r} \mathbb{E}_{\{\tilde{x}_r^k\}} \left[ w_r^k \log p_\theta(\tilde{x}_r^k) \right]$ for the root and $\theta_i^{n+1} = \arg\max_{\theta_i} \mathbb{E}_{\{\tilde{x}_i^k, \tilde{x}_{\mathrm{p}(i)}^k\}} \left[ w_i^k \log p_\theta(\tilde{x}_i^k \mid \tilde{x}_{\mathrm{p}(i)}^k) \right]$ for non-root nodes. These updates can be performed separately, as in the fully-decomposed EDA, but are coupled according to the junction tree by the weights.

For a formal derivation of DADO, see Sec. 2.1 or Sec. A.4.

## A.4 DADO DERIVATION

Herein, we give an augmented version of the derivation presented in Sec. 2.1. We build on the derivation of the naive EDA given in Sec. A.2. The goal of DADO is to obtain a generative model, $p_\theta(x)$, that maximizes the EDA objective, $\arg\max_\theta \mathbb{E}_{p_\theta(x)}[f(x)]$, while leveraging decomposability in $f(x)$ for optimization efficiency. DADO requires as input a decomposed version of $f(x)$, which can be described by an undirected junction tree, $\mathcal{T} := (\mathcal{N}, \mathcal{E})$, with nodes, $\mathcal{N}$, and edges, $\mathcal{E}$. As noted in the introduction and shown in our experiments on proteins, identifying useful decomposability is feasible in practice. Given the junction tree decomposition, we write $f(x) = \sum_{i \in \mathcal{N}} f_i(\tilde{x}_i) + \sum_{(i,j) \in \mathcal{E}} f_{i,j}(\tilde{x}_i, \tilde{x}_j)$, where we refer to $f_i(\tilde{x}_i)$ as "node component functions" and $f_{i,j}(\tilde{x}_i, \tilde{x}_j)$ as "edge component functions" (Fig. 1a). This functional structure is intimately related to notions of the "epistatic landscape" of a protein's property function and spectral analysis (see Sec. A.1.1). When the component functions are not known *a priori*, they can be parameterized and fit to labeled data.

We will begin our exposition by recalling how to do decomposition-aware exact (non-distributional) optimization, that is, to solve $\arg\max_x f(x)$. This problem is efficiently solved with a classical message-passing algorithm, which coordinates local optimizations across parts of the junction tree to obtain a single global optimum. Its efficiency comes from breaking optimization over all variables jointly into separate optimizations for each (smaller) meta-variable. Having loaded the reader with this intuition, we then adapt these ideas to distributional optimization, yielding DADO.

### A.4.1 CLASSICAL MESSAGE-PASSING FOR NON-DISTRIBUTIONAL OPTIMIZATION

Although classical message-passing has been largely used for probabilistic inference on probabilistic graphical models (Pearl, 1988; Shah, 2014), it can also be used for exact optimization of a function, $x^* = \arg\max_x f(x)$ (Vlassis et al., 2004). In particular, message-passing can be used to find a global maximum of a function defined on an undirected junction tree, $\mathcal{T}$, by making use of its topology for optimal time-complexity. In particular, one takes the junction tree, which is undirected, and roots it to obtain a directed tree, which induces a hierarchy among the meta-variables from root to leaves. Each node is responsible for accumulating information from all nodes in its sub-tree and then passing this information on to its parent. Consequently, the root node receives information from the entire tree, which is sufficient to set its variables in a globally optimal manner. Then, starting with the root, each parent communicates its variables' optimal settings to its children, which can in turn set their variables optimally, and so forth.

To obtain the rooted tree, $\mathcal{T}' := (\mathcal{N}', \mathcal{E}')$, from the junction tree, one keeps the same nodes, $\mathcal{N}' = \mathcal{N}$, chooses a root node, $r$, and directs all edges in $\mathcal{E}$ outward from $r$, yielding directed edges, $\mathcal{E}'$. Although rooting at any node will suffice, we choose $r$ such that $\mathcal{T}'$ has the shortest height possible. Message-passing finds a global optimum in two passes through the tree: one round of passing messages up from leaves to root, and then one round passing messages back down to the leaves.

Given $\mathcal{T}'$, classical message-passing first uses dynamic programming to accumulate information from the leaves up to the root. Similarly to any dynamic programming procedure, we accumulate solutions to increasingly larger intermediate sub-problems. In this case, each sub-problem is to find the value of $f(x)$ evaluated on only a subset of meta-variables, rather than on the full set of variables in $x$. Each sub-problem is tractable owing to the decomposition of the objective function into component functions for each node and edge, and by respecting the partial order of sub-problems induced by $\mathcal{T}'$. Specifically, one computes a *value function*, $V_i^{\max}(\tilde{x}_{\mathrm{p}(i)})$, for each node $i \in \mathcal{N}' \setminus \{r\}$, which tells us for each setting of its parent, $\tilde{x}_{\mathrm{p}(i)}$, the value of the intermediate objective function defined by the edge component function, $f_{\mathrm{p}(i),i}(\tilde{x}_{\mathrm{p}(i)}, \tilde{x}_i)$, plus all component functions over the sub-tree rooted at $i$, given that all nodes maximize their respective intermediate objectives. Computing value functions comprises the first pass through the tree, from leaves to root:

$$V_i^{\max}(\tilde{x}_{\mathrm{p}(i)}) := \max_{\tilde{x}_i} \left( f_i(\tilde{x}_i) + f_{\mathrm{p}(i),i}(\tilde{x}_{\mathrm{p}(i)}, \tilde{x}_i) + \sum_{c \in \mathrm{children}(i)} V_c^{\max}(\tilde{x}_i) \right).$$

Notably, $V_i^{\max}(\tilde{x}_{\mathrm{p}(i)})$ provides sufficient information about all nodes in the sub-tree rooted at $i$ to optimally choose the value of $\tilde{x}_{\mathrm{p}(i)}$ with respect to its children. Thus it follows that once all value functions have been computed, the root's assignment can be set in a globally optimal manner from

its children's value functions,

$$\tilde{x}_r^* := \arg\max_{\tilde{x}_r} \left( f_r(\tilde{x}_r) + \sum_{c \in \text{children}(r)} V_c(\tilde{x}_r) \right).$$

Having chosen the root assignment, we then pass it down the tree as $\tilde{x}_{\text{p}(i)} = \tilde{x}_{\text{p}(i)}^*$ to its children, which successively pass their chosen assignments to their children, all the way to the leaves,

$$\tilde{x}_i^* := \arg\max_{\tilde{x}_i} \left( f_i(\tilde{x}_i) + f_{\text{p}(i),i}(\tilde{x}_{\text{p}(i)}^*, \tilde{x}_i) + \sum_{c \in \text{children}(i)} V_c(\tilde{x}_i) \right),$$

resulting in a global maximizer $x^*$ of $f(x)$. This dynamic programming "traceback" of optimal assignments back down the tree constitutes our second and final pass of messages.

**Alternative notation.** For convenience of our generalization to distributional optimization, we re-write the parent value functions $V_i^{\text{max}}(\tilde{x}_{\text{p}(i)})$ in terms of child-parent, $Q_i^{\text{max}}(\tilde{x}_i, \tilde{x}_{\text{p}(i)})$, and single-node, $Q_i^{\text{max}}(\tilde{x}_i)$ value functions:

$$V_i^{\text{max}}(\tilde{x}_{\text{p}(i)}) := \max_{\tilde{x}_i} \ Q_i^{\text{max}}(\tilde{x}_i, \tilde{x}_{\text{p}(i)}), \quad \text{where} \tag{9}$$

$$Q_i^{\text{max}}(\tilde{x}_i, \tilde{x}_{\text{p}(i)}) := f_{\text{p}(i),i}(\tilde{x}_{\text{p}(i)}, \tilde{x}_i) + Q_i^{\text{max}}(\tilde{x}_i) \quad \text{and} \tag{10}$$

$$Q_i^{\text{max}}(\tilde{x}_i) := f_i(\tilde{x}_i) + \sum_{c \in \text{children}(i)} V_c^{\text{max}}(\tilde{x}_i). \tag{11}$$

In contrast to the original parent value functions, $Q_i^{\text{max}}(\tilde{x}_i, \tilde{x}_{\text{p}(i)})$ represents the effect of the choice of *both* $\tilde{x}_{\text{p}(i)}$ and $\tilde{x}_i$ on their edge component function plus all component functions over the sub-tree rooted at $i$, assuming all descendants of $i$ maximize their corresponding value functions. Intuitively, $Q_i^{\text{max}}(\tilde{x}_i, \tilde{x}_{\text{p}(i)})$ is the value function on edge $(\text{p}(i), i)$ prior to $\tilde{x}_i$ being maximized out, which will become useful if we want to, say, sample $\tilde{x}_i$ according to some distribution instead. $Q_i^{\text{max}}(\tilde{x}_i, \tilde{x}_{\text{p}(i)})$ is composed of two terms, one of which depends on its parent, and one of which doesn't, $Q_i^{\text{max}}(\tilde{x}_i)$. Written using the $Q$-functions just defined, the equivalent traceback equations for selecting a globally optimal assignment, $x^*$, are simply $\tilde{x}_r^* := \arg\max_{\tilde{x}_r} Q_r^{\text{max}}(\tilde{x}_r)$ and $\tilde{x}_i^* := \arg\max_{\tilde{x}_i} Q_i^{\text{max}}(\tilde{x}_i, \tilde{x}_{\text{p}(i)}^*)$. In other words,

$$x^* = \arg\max_x f(x) = \{\arg\max_{\tilde{x}_r} Q_r^{\text{max}}(\tilde{x}_r)\} \ + \ \{\arg\max_{\tilde{x}_i} Q_i^{\text{max}}(\tilde{x}_i, \tilde{x}_{\text{p}(i)}^*)\}_{(\text{p}(i),i) \in \mathcal{E}'} \tag{12}$$

$$= \arg\max_x \left( Q_r^{\text{max}}(\tilde{x}_r) + \sum_{(\text{p}(i),i) \in \mathcal{E}'} Q_i^{\text{max}}(\tilde{x}_i, \tilde{x}_{\text{p}(i)}) \right). \tag{13}$$

### A.4.2 FROM CLASSICAL MESSAGE PASSING TO DISTRIBUTIONAL OPTIMIZATION

In the same way that an EDA transforms $\arg\max_x f(x)$ into a distributional optimization problem (Sec. A.2), we can rewrite the equivalent message-passing objective in Eq. 13 as DO. Because the original optimization problems over $x$ are equivalent, their DO formulations are equivalent too,

$$\arg\max_\theta \mathbb{E}_{p_\theta(x)}[f(x)] = \arg\max_\theta \mathbb{E}_{p_\theta(x)} \left[ Q_r^{\text{max}}(\tilde{x}_r) + \sum_{(\text{p}(i),i) \in \mathcal{E}'} Q_i^{\text{max}}(\tilde{x}_i, \tilde{x}_{\text{p}(i)}) \right] \tag{14}$$

$$= \arg\max_\theta \left( \mathbb{E}_{p_\theta(x)}[Q_r^{\text{max}}(\tilde{x}_r)] + \sum_{(\text{p}(i),i) \in \mathcal{E}'} \mathbb{E}_{p_\theta(x)}[Q_i^{\text{max}}(\tilde{x}_i, \tilde{x}_{\text{p}(i)})] \right), \tag{15}$$

where we've used linearity of expectations. However, a generic joint search distribution cannot take advantage of the linear additivity in value functions over the junction tree topology. That is, while the classical traceback equations perform maximization over each meta-variable separately, Eq. 15 uses a single, unfactorized search distribution over *all* variables, $p_\theta(x)$, to optimize each $Q$-function. We address this next, by factorizing the search distribution.

**Factorized search distribution.** Classical message-passing (Eq. 12) independently maximizes each $Q$-function conditional on the choice of $\tilde{x}_{\text{p}(i)}$, instead of explicitly maximizing all variables in $x$ jointly (Eq. 13). This is possible because each $Q_i$ captures all relevant global information needed to choose $\tilde{x}_i$. It stands to reason then that DO can do something similar. Specifically, instead of training a single joint search distribution, we train smaller search distributions over each $\tilde{x}_i$, conditional on $\tilde{x}_{\text{p}(i)}$, to separately maximize each $Q_i^{\text{max}}(\tilde{x}_i, \tilde{x}_{\text{p}(i)})$. That is, we factor the search distribution according to $\mathcal{T}'$, resulting in a DAG search distribution, $p_\theta(x) := p_\theta(\tilde{x}_r) \prod_{(\text{p}(i),i) \in \mathcal{E}'} p_\theta(\tilde{x}_i \mid \tilde{x}_{\text{p}(i)})$,

for root node $r$, non-root nodes $i$, and parents $\mathrm{p}(i)$, connected by directed edges $\mathcal{E}'$, and with parameters, $\theta$. This factorized search distribution can be plugged into Eq. 15 for an equivalent optimization problem. We refer to each element of this product as one of the *factors* of the search distribution. In our implementation, each factor has completely separate parameters, though we write a shared $\theta$ for conciseness. Notice that each factor of the search distribution interacts with each other factor through the directed edges, $\mathcal{E}'$. That is, the distribution of $\tilde{x}_i$ depends on its parent's factor, and through it, all of its parent's ancestors: $p_\theta(\tilde{x}_i) = p_\theta(\tilde{x}_i \mid \tilde{x}_{\mathrm{p}(i)}) p_\theta(\tilde{x}_{\mathrm{p}(i)})$. In turn, each of node $i$'s children's factors depends on $p_\theta(\tilde{x}_i)$. Due to this coupling, we cannot optimize each factor fully independently. But we can still *update* each factor separately from the others in a globally-consistent manner via message-passing. In particular, each factor will only be responsible for directly optimizing its own meta-variable, but will need to coordinate with its neighboring factors by getting information from them about how they are optimizing their meta-variables in a manner analogous to classical message-passing. Our current messages, $Q_i^{\max}$, convey the value of each intermediate objective when all meta-variables are chosen via maximization. For DO, we'll require messages that communicate values when meta-variables are chosen according to the factorized search distribution, $p_\theta(x)$.

**Distributional value functions.** While one certainly could choose to optimize classical value functions using an EDA search distribution (Eq. 15), it doesn't make sense for two main reasons. As we just mentioned, DO aims to train $p_\theta(x)$ such that it maximizes $f(x)$, or equivalently, the sum of $Q$-functions, in expectation. Therefore, the DO objective should consider intermediate objective values for meta-variables chosen according to the current search distribution, not those chosen by explicit maximization, as in $V_i^{\max}(\tilde{x}_{\mathrm{p}(i)})$. Additionally, computing classical value functions requires enumerating all assignments of each $\tilde{x}_i$ for the maximum used in $V_i^{\max}(\tilde{x}_{\mathrm{p}(i)})$. When $\tilde{x}_i$ contains more than a few design variables, this `max` operation quickly becomes intractable. One reason for doing distributional optimization is to avoid enumerating massive design spaces. To address both issues, we define corresponding *distributional* value functions that fulfill both desiderata:

$$V_i^\theta(\tilde{x}_{\mathrm{p}(i)}) := \mathbb{E}_{p_\theta(\tilde{x}_i \mid \tilde{x}_{\mathrm{p}(i)})}[Q_i^\theta(\tilde{x}_i, \tilde{x}_{\mathrm{p}(i)})], \quad \text{where} \tag{16}$$

$$Q_i^\theta(\tilde{x}_i, \tilde{x}_{\mathrm{p}(i)}) := f_{\mathrm{p}(i),i}(\tilde{x}_{\mathrm{p}(i)}, \tilde{x}_i) + Q_i^\theta(\tilde{x}_i) \quad \text{and} \tag{17}$$

$$Q_i^\theta(\tilde{x}_i) := f_i(\tilde{x}_i) + \sum_{c \in \mathrm{children}(i)} V_c^\theta(\tilde{x}_i). \tag{18}$$

Compared to Eq. 9, the distributional $V$-functions compute the value in expectation under $p_\theta$ instead of a `max`. Because they use an expectation instead of a `max` operation, these value functions can be approximated tractably and without bias by drawing Monte-Carlo samples from the search distribution. The connection to classical value functions can be made clearer by defining a particular distribution which recovers them, the one placing all of its mass on the `arg max`, and its corresponding distributional value functions:

$$p_{\max}(\tilde{x}_i = A \mid \tilde{x}_{\mathrm{p}(i)}) = \begin{cases} 1 & \text{if } A = \arg\max_{\tilde{x}_i} Q_i^{\max}(\tilde{x}_i, \tilde{x}_{\mathrm{p}(i)}) \\ 0 & \text{otherwise} \end{cases}, \tag{19}$$

$$V_i^{\max}(\tilde{x}_{\mathrm{p}(i)}) = \mathbb{E}_{p_{\max}(\tilde{x}_i \mid \tilde{x}_{\mathrm{p}(i)})}[Q_i^{\max}(\tilde{x}_i, \tilde{x}_{\mathrm{p}(i)})] = \max_{\tilde{x}_i} f_i(\tilde{x}_i) + f_{\mathrm{p}(i),i}(\tilde{x}_{\mathrm{p}(i)}, \tilde{x}_i) + \sum_{c \in \mathrm{children}(i)} V_c^{\max}(\tilde{x}_i), \tag{20}$$

$$Q_i^{\max}(\tilde{x}_i, \tilde{x}_{\mathrm{p}(i)}) = f_{\mathrm{p}(i),i}(\tilde{x}_{\mathrm{p}(i)}, \tilde{x}_i) + Q_i^{\max}(\tilde{x}_i), \quad \text{for } Q_i^{\max}(\tilde{x}_i) = f_i(\tilde{x}_i) + \sum_{c \in \mathrm{children}(i)} V_c^{\max}(\tilde{x}_i) \tag{21}$$

Moreover, each distributional value function lower bounds each corresponding classical value function because the expectation of a function cannot exceed its maximum. This relation is evident for the base case of a leaf node,

$$V_i^{\max}(\tilde{x}_{\mathrm{p}(i)}) = \max_{\tilde{x}_i} f_i(\tilde{x}_i) + f_{\mathrm{p}(i),i}(\tilde{x}_{\mathrm{p}(i)}, \tilde{x}_i) \geq \mathbb{E}_{p_\theta(\tilde{x}_i \mid \tilde{x}_{\mathrm{p}(i)})}[f_i(\tilde{x}_i) + f_{\mathrm{p}(i),i}(\tilde{x}_{\mathrm{p}(i)}, \tilde{x}_i)] = V_i^\theta(\tilde{x}_{\mathrm{p}(i)}), \tag{22}$$

and by inductive argument, is also true for all other nodes' $V$-functions, and similarly, the $Q$-functions, which only differ in which $V$-functions they sum over. This implies that the sum of

classical value functions in the objective in Eq. 15 is bounded below by the sum of distributional value functions. This bound still holds when we take the expectation of each sum under $p_\theta(x)$,

$$\mathbb{E}_{p_\theta(x)}[Q_r^{\max}(\tilde{x}_r)] + \sum_{(\mathrm{p}(i),i)\in\mathcal{E}'} \mathbb{E}_{p_\theta(x)}[Q_i^{\max}(\tilde{x}_i,\tilde{x}_{\mathrm{p}(i)})] \geq \mathbb{E}_{p_\theta(x)}[Q_r^\theta(\tilde{x}_r)] + \sum_{(\mathrm{p}(i),i)\in\mathcal{E}'} \mathbb{E}_{p_\theta(x)}[Q_i^\theta(\tilde{x}_i,\tilde{x}_{\mathrm{p}(i)})].$$
(23)

We'll optimize the lower bound with its distributional value functions instead of the sum of classical value functions.

**Distributional optimization with value functions.** All that remains is to derive an update rule that treats each search distribution factor separately. Since each expectand in Eq. 23 doesn't depend on descendant meta-variables, we can replace $p_\theta(x)$ with each meta-variable's marginal distribution:

$$\arg\max_\theta \mathbb{E}_{p_\theta(\tilde{x}_r)}[Q_r^\theta(\tilde{x}_r)] + \sum_{(\mathrm{p}(i),i)\in\mathcal{E}'} \mathbb{E}_{p_\theta(\tilde{x}_i|\tilde{x}_{\mathrm{p}(i)})p_\theta(\tilde{x}_{\mathrm{p}(i)})}[Q_i^\theta(\tilde{x}_i,\tilde{x}_{\mathrm{p}(i)})].$$
(24)

We then follow the EDA derivation (Sec. A.2) to arrive at a sample-approximated update rule using the previous iteration's distribution, $p_{\theta^n}(x)$ in which each term is optimized by only a single factor. We omit the derivation for the root factor because it'ss exactly the same as in Sec. A.2, using $Q_r^{\theta^n}(\tilde{x}_r)$ as the weight instead of $f(x)$. To update a conditional factor from Eq. 24 in an EDA loop, we define $p_n^*(\tilde{x}_c \mid \tilde{x}_{\mathrm{p}(i)}) \propto Q_c^{\theta^n}(\tilde{x}_c,\tilde{x}_{\mathrm{p}(i)}) \cdot p_{\theta^n}(\tilde{x}_c \mid \tilde{x}_{\mathrm{p}(i)})$, and minimize its divergence from $p^\theta(\tilde{x}_c \mid \tilde{x}_{\mathrm{p}(i)})$ using the law of iterated expectations:

$$-\mathbb{E}_{p_{\theta^n}(\tilde{x}_{\mathrm{p}(i)})}\left[D_{\mathrm{KL}}\left(p_n^*(\tilde{x}_c \mid \tilde{x}_{\mathrm{p}(i)}) \,\|\, p_\theta(\tilde{x}_c \mid \tilde{x}_{\mathrm{p}(i)})\right)\right]$$
(25)

$$= \frac{1}{Z}\mathbb{E}_{p_{\theta^n}(\tilde{x}_{\mathrm{p}(i)})}\left[\mathbb{E}_{p_{\theta^n}(\tilde{x}_c|\tilde{x}_{\mathrm{p}(i)})}[Q_c^{\theta^n}(\tilde{x}_c,\tilde{x}_{\mathrm{p}(i)})\log p_\theta(\tilde{x}_c \mid \tilde{x}_{\mathrm{p}(i)})] + H(p_n^*(\tilde{x}_c \mid \tilde{x}_{\mathrm{p}(i)}))\right].$$
(26)

Again, when taking the $\arg\max$ with respect to $\theta$, we can equivalently remove the entropy term which bears no dependence on $\theta$ and division by constant $Z$. Notice that the dependence of the conditional terms on $p_\theta(\tilde{x}_{\mathrm{p}(i)})$ in Eq. 24 is approximated by sampling, which is why only individual factors of the search distribution, $\log p_\theta(\tilde{x}_c \mid \tilde{x}_{\mathrm{p}(i)})$, appear in each summand, as opposed to $\log p_\theta(\tilde{x}_c \mid \tilde{x}_{\mathrm{p}(i)})p_\theta(\tilde{x}_{\mathrm{p}(i)})$. Updating $p_\theta(\tilde{x}_{\mathrm{p}(i)})$ to maximize $Q_c^\theta(\tilde{x}_c,\tilde{x}_{\mathrm{p}(i)})$ would be redundant and against the spirit of message-passing since parent $Q$-functions already include that info from children. The resulting update rule over all search distribution factors (sharing a single set of samples; see Fig. 1b) can be written as a sum of weighted likelihoods for each factor,

$$\theta^{n+1} = \arg\max_\theta \sum_{x\sim p_{\theta^n}(x)} \left(Q_r^{\theta^n}(\tilde{x}_r)\log p_\theta(\tilde{x}_r) + \sum_{(\mathrm{p}(i),i)\in\mathcal{E}'} Q_i^{\theta^n}(\tilde{x}_i,\tilde{x}_{\mathrm{p}(i)})\log p_\theta(\tilde{x}_i \mid \tilde{x}_{\mathrm{p}(i)})\right),$$
(27)

which can be written equivalently as separate updates because the factors don't share parameters, as desired:

$$\theta_r^{n+1} = \arg\max_{\theta_r} \sum_{x\sim p_{\theta^n}(x)} Q_r^{\theta^n}(\tilde{x}_r)\log p_{\theta_r}(\tilde{x}_r), \quad \text{and}$$
(28)

$$\theta_i^{n+1} = \arg\max_{\theta_i} \sum_{x\sim p_{\theta^n}(x)} Q_i^{\theta^n}(\tilde{x}_i,\tilde{x}_{\mathrm{p}(i)})\log p_{\theta_i}(\tilde{x}_i \mid \tilde{x}_{\mathrm{p}(i)}), \quad \forall\, i\in\mathcal{N}\setminus\{r\}.$$
(29)

Each factor is weighted by its corresponding value function, enabling it to coordinate with all its descendant factors despite their being updated separately. The whole DO is tied together at the top by the root factor. Our resulting algorithm (Alg. 1) fits into the three EDA steps: (1) designs are sampled from $p_{\theta^n}(x)$, (2) weights, here each $Q$-function instead of $f(x)$, are computed, and (3), each search distribution factor receives its own independent weighted maximum likelihood update based on these weights (steps 1 and 2 illustrated in Fig. 1b). These are repeated until convergence, or for some fixed number of iterations. The factor updates are coupled only through the $Q$-functions, the messages across edges. $\{Q_i^{\theta^n}\}_{i\in\mathcal{N}}$ are only valid while $\theta$ is close to $\theta^n$, meaning one must balance how many gradient steps are taken before drawing new samples. If too many gradient steps are taken without resampling, a factor may be changing its parameters to collaborate with another factor which has already changed its behavior. It's common for EDAs to include an additional hyperparameter $W(\cdot)$, a monotonic shaping function applied to the weights, to alter optimization dynamics (Brookes et al., 2020), so we include it in Alg.1( 1 on lines 16 and 18. Choosing $W$ to be the identity recovers our derivation above. Notice that in place of the classical message-passing traceback equations, DADO simply performs sequential conditional sampling from its search distribution. Interestingly, a very similar algorithm can be derived without using a lower bound on the value functions (Eq. 23) if one adds an entropy-maximizing term to the initial DO objective (Sec. A.5).

## A.5 ALTERNATIVE DADO DERIVATION: MAXIMUM ENTROPY OBJECTIVE

If, by comparison to Sec. A.4, we instead seek to solve a closely related objective which additionally maximizes the search distribution's entropy, we can arrive at a similar method by leveraging the probabilistic graphical modeling (PGM) framework in Levine (2018). This view will provide another intuition for the relationship between classical value functions and distributional value functions approximated with Monte-Carlo samples. It also highlights the core components of our conceptual framework, which are shared even by derivations starting from different objectives. In other words, one need not use our specific method as outlined in Alg. 1 to reap the benefits of decomposition-aware distributional optimization, and variations may have desirable properties and/or simply perform better in certain scientific design settings. *This derivation will assume that you've already read key parts of Sec. 2/Sec. A.4 and are familiar with the function decomposition, search distribution factorization, and value functions.*

**Maximum entropy decomposition-aware distributional optimization.** We begin from the maximum entropy problem (Jaynes, 1957; Ziebart et al., 2008; Zhu et al., 2024),

$$\arg\max_\theta \mathbb{E}_{p_\theta(x)}[f(x)] + \beta H(p_\theta), \quad \beta \geq 0. \tag{30}$$

This objective is arguably more consistent with the desired end result for many scientific design problems. Often, a distribution over good solutions is preferred to a single solution (*i.e.*, a distribution with no entropy)[1]. When optimizing a predictive model or more broadly, dealing with uncertainty or inaccuracy in the objective function, having a distribution over good solutions can be particularly important. The maximum entropy objective has a closed-form solution, $p^*(x) \propto \exp f(x)/\beta$, motivating the solution of an equivalent variational objective, $\arg\min_\theta D_{\mathrm{KL}}(p_\theta \,\|\, p^*)$, from which we will obtain a decomposed update rule. We first plug in decomposed versions of $f$ and $p_\theta(x)$, as described in Sec. 2.1/Sec. A.4, to get

$$\arg\min_\theta D_{\mathrm{KL}}(p_\theta \,\|\, p^*) = \arg\min_\theta D_{\mathrm{KL}}\left(p_\theta(\tilde{x}_r) \prod_{i\in\mathcal{N}\setminus\{r\}} p_\theta(x_i \mid \tilde{x}_{\mathrm{p}(i)}) \,\|\, \exp\frac{\sum_{i\in\mathcal{N}} f_{i,p}(\tilde{x}_i, \tilde{x}_{\mathrm{p}(i)}) + f_i(\tilde{x}_i)}{\beta}\right). \tag{31}$$

Notice that we've dropped the normalizing constant for $p^*$ because it has no dependence on $\theta$. In what follows, for compactness and clarity, we will abuse notation slightly by including the root factor, $p_\theta(\tilde{x}_r)$, with the other nodes, writing it as conditional on a parent even though it has no parent and is an unconditional distribution.

$$-D_{\mathrm{KL}}\left(\prod_{i\in\mathcal{N}} p_\theta(x_i \mid \tilde{x}_{\mathrm{p}(i)}) \,\|\, \exp\frac{\sum_{i\in\mathcal{N}} f_{i,p}(\tilde{x}_i, \tilde{x}_{\mathrm{p}(i)}) + f_i(\tilde{x}_i)}{\beta}\right) \tag{32}$$

$$= \mathbb{E}_{p_\theta(x)}\left[\frac{\sum_{i\in\mathcal{N}} f_{i,p}(\tilde{x}_i, \tilde{x}_{\mathrm{p}(i)}) + f_i(\tilde{x}_{\mathrm{p}(i)})}{\beta} - \log\prod_{i\in\mathcal{N}} p_\theta(x_i \mid \tilde{x}_{\mathrm{p}(i)})\right] \tag{33}$$

$$= \mathbb{E}_{p_\theta(x)}\left[\sum_{i\in\mathcal{N}} \frac{f_{i,p}(\tilde{x}_i, \tilde{x}_{\mathrm{p}(i)}) + f_i(\tilde{x}_{\mathrm{p}(i)})}{\beta} - \log p_\theta(\tilde{x}_i \mid \tilde{x}_{\mathrm{p}(i)})\right] \tag{34}$$

$$= \sum_{i\in\mathcal{N}} \mathbb{E}_{p_\theta(\tilde{x}_i, \tilde{x}_{\mathrm{p}(i)})}\left[\frac{f_{i,p}(\tilde{x}_i, \tilde{x}_{\mathrm{p}(i)}) + f_i(\tilde{x}_{\mathrm{p}(i)})}{\beta} - \log p_\theta(\tilde{x}_i \mid \tilde{x}_{\mathrm{p}(i)})\right] \tag{35}$$

The final line decomposes over nodes, but still has an explicit dependence on all ancestor node search distribution factors. One way to obtain a separate update for each search distribution factor is to derive a form of $p^*(x)$ factorized in accordance with $p_\theta(x)$'s factorization, so that we can simply minimize each factor's divergence from the optimal one.

---

[1]In our experiments (no entropy bonus), the search distribution avoids collapse because we don't run until convergence (fixed number of iterations), and because we sweep the learning rate (if it's very high, the distribution may quickly collapse to a point, limiting further improvement). Similar strategies were used by Brookes et al. (2019) instead of an explicit entropy bonus. EDAs were originally used for discrete optimization problems where a single solution was desired. Using distributional optimization to solve more nuanced problems calls for some modifications.

**Optimal search distribution factors via message-passing.** To derive the optimal factors, $p^*(\tilde{x}_i \mid \tilde{x}_{\mathrm{p}(i)})$, we'll use message-passing, for which we now introduce some notation. Note that we adopt the same convention as above of including the root with the rest of the value functions, meaning writing it with a parent, although it doesn't have a parent and all terms involving its nonexistent parent are simply 0. We recursively define value functions (consistent with Levine (2018)) as:

$$V_i(\tilde{x}_{\mathrm{p}(i)}) = \log \sum\nolimits_{\tilde{x}_i} \exp Q_i(\tilde{x}_i, \tilde{x}_{\mathrm{p}(i)}), \tag{36}$$

$$Q_i(\tilde{x}_i, \tilde{x}_{\mathrm{p}(i)}) = f_{\mathrm{p}(i),i}(\tilde{x}_{\mathrm{p}(i)}, \tilde{x}_i) + Q_i(\tilde{x}_i), \text{ and } Q_i(\tilde{x}_i) = f_i(\tilde{x}_i) + \sum\nolimits_{c \in \mathrm{children}(i)} V_c(\tilde{x}_i). \tag{37}$$

Though they might seem a bit arbitrary, substituting the value functions into Eq. 35 will help illuminate $p^*(\tilde{x}_i \mid \tilde{x}_{\mathrm{p}(i)})$. Notice that the $Q$-functions have the same recursive definition as in DADO, though the $V$-functions are different. We first consider the base case of a leaf node, where $Q_i(\tilde{x}_i, \tilde{x}_{\mathrm{p}(i)}) = f_{\mathrm{p}(i),i}(\tilde{x}_{\mathrm{p}(i)}, \tilde{x}_i) + f_i(\tilde{x}_i)$. We substitute, add $0 = V_i(\tilde{x}_{\mathrm{p}(i)})/\beta - V_i(\tilde{x}_{\mathrm{p}(i)})/\beta$, and use the definition of KL divergence:

$$\mathbb{E}_{p_\theta(\tilde{x}_i \mid \tilde{x}_{\mathrm{p}(i)}) p_\theta(\tilde{x}_{\mathrm{p}(i)})} \left[ \log \exp \frac{Q_i(\tilde{x}_i, \tilde{x}_{\mathrm{p}(i)})}{\beta} - \log p_\theta(\tilde{x}_i \mid \tilde{x}_{\mathrm{p}(i)}) + \frac{V_i(\tilde{x}_{\mathrm{p}(i)})}{\beta} - \log \exp \frac{V_i(\tilde{x}_{\mathrm{p}(i)})}{\beta} \right] \tag{38}$$

$$= \mathbb{E}_{p_\theta(\tilde{x}_{\mathrm{p}(i)})} \left[ -D_{\mathrm{KL}}\left( p_\theta(\tilde{x}_i \mid \tilde{x}_{\mathrm{p}(i)}) \,\|\, \exp \frac{Q_i(\tilde{x}_i, \tilde{x}_{\mathrm{p}(i)}) - V_i(\tilde{x}_{\mathrm{p}(i)})}{\beta} \right) + \frac{V_i(\tilde{x}_{\mathrm{p}(i)})}{\beta} \right]. \tag{39}$$

Because $p_\theta(\tilde{x}_i \mid \tilde{x}_{\mathrm{p}(i)})$ only appears in the KL divergence, the overall expectation will be maximized, as far as $p_\theta(\tilde{x}_i \mid \tilde{x}_{\mathrm{p}(i)})$ is concerned, when the divergence is 0. Therefore, for a leaf node, $p^*(\tilde{x}_i \mid \tilde{x}_{\mathrm{p}(i)}) = \exp((Q_i(\tilde{x}_i, \tilde{x}_{\mathrm{p}(i)}) - V_i(\tilde{x}_{\mathrm{p}(i)}))/\beta)$, which is a proper distribution because $\exp V_i(\tilde{x}_{\mathrm{p}(i)})$ is exactly the normalizing constant of $\exp Q_i(\tilde{x}_i, \tilde{x}_{\mathrm{p}(i)})$. Notice that we still have a term, $\mathbb{E}_{p_\theta(\tilde{x}_{\mathrm{p}(i)})} \left[ V_i(\tilde{x}_{\mathrm{p}(i)})/\beta \right]$, outside of the divergence (Eq. 39). Maximizing this term must therefore be the responsibility of $p_\theta(\tilde{x}_{\mathrm{p}(i)})$, not $p_\theta(\tilde{x}_i \mid \tilde{x}_{\mathrm{p}(i)})$. In fact, these leftover terms are included in the parent's $Q$-function via the sum over all child $V$-functions in $Q_i(\tilde{x}_i)$ as we defined it above (Eq. 37). It follows that the recursive case has the same form as the leaves, which had $Q_i(\tilde{x}_i) = f_i(\tilde{x}_i)$ because they've no children (more detailed derivation in Levine (2018)). That is, for all non-root nodes $i$, $p^*(\tilde{x}_i \mid \tilde{x}_{\mathrm{p}(i)}) = \exp((Q_i(\tilde{x}_i, \tilde{x}_{\mathrm{p}(i)}) - V_i(\tilde{x}_{\mathrm{p}(i)}))/\beta)$; for the root node, $p^*(\tilde{x}_r) = \exp((Q_r(\tilde{x}_r) - V_r)/\beta)$, where $V_r = \log \sum_{\tilde{x}_r} \exp Q_r(\tilde{x}_r)$ is a normalizing constant with no dependence on $\theta$. We now have an optimal form for each factor and can proceed to minimize our search factors' divergences from them completely separately.

**Distributional optimization by matching optimal factors.** Given a factorization of the optimal search distribution, $p^*(x)$, we can train $p_\theta(x)$'s factors to match their corresponding optimal factor distributions, resulting in an algorithm very similar to DADO. The objective for node $i$'s factor minimizes its divergence to the optimal factor $p^*(\tilde{x}_i \mid \tilde{x}_{\mathrm{p}(i)})$,

$$\arg\min_{\theta_i} \mathbb{E}_{p_{\theta_{\mathrm{p}(i)}}(\tilde{x}_{\mathrm{p}(i)})} \left[ D_{\mathrm{KL}}\left( p_{\theta_i}(\tilde{x}_i \mid \tilde{x}_{\mathrm{p}(i)}) \,\|\, \exp \frac{Q_i(\tilde{x}_i, \tilde{x}_{\mathrm{p}(i)}) - V_i(\tilde{x}_{\mathrm{p}(i)})}{\beta} \right) \right] \tag{40}$$

$$= \arg\max_{\theta_i} \mathbb{E}_{p_{\theta_{\mathrm{p}(i)}}(\tilde{x}_{\mathrm{p}(i)})} \left[ \mathbb{E}_{p_{\theta_i}(\tilde{x}_i \mid \tilde{x}_{\mathrm{p}(i)})} \left[ Q_i(\tilde{x}_i, \tilde{x}_{\mathrm{p}(i)}) \right] - V_i(\tilde{x}_{\mathrm{p}(i)}) + \beta H(p_{\theta_i}(\tilde{x}_i \mid \tilde{x}_{\mathrm{p}(i)})) \right], \tag{41}$$

and an analogous objective can be written for the root node. Notice that $V_i(\tilde{x}_{\mathrm{p}(i)})$ can be dropped completely or set to any other constant (with respect to $\tilde{x}_i$) without altering the $\arg\max$, though certain settings may yield more favorable optimization dynamics [2]. We approximate the expectations in the objective using samples from the current search distribution (as in EDAs) to yield a weighted maximum likelihood update rule for each node $i$,

$$\theta_i^{n+1} = \arg\max_{\theta_i} \sum_{x \sim p_{\theta^n}(x)} \left( \left( Q_i(\tilde{x}_i, \tilde{x}_{\mathrm{p}(i)}) - V_i(\tilde{x}_{\mathrm{p}(i)}) \right) \log p_{\theta_i}(\tilde{x}_i \mid \tilde{x}_{\mathrm{p}(i)}) - \beta \log p_{\theta_i}(\tilde{x}_i \mid \tilde{x}_{\mathrm{p}(i)}) \right), \tag{42}$$

---

[2] $V_i(\tilde{x}_{\mathrm{p}(i)})$ is often thought of as a "baseline" in the RL literature, the choice of which doesn't bias the objective's gradient, but can reduce its variance substantially and lead to more efficient policy optimization (Sutton et al., 1998).

and since the weight should always be non-negative, a monotonic shaping function, $W$, outputting non-negative weights, can be introduced too,

$$\theta_i^{n+1} = \arg\max_{\theta_i} \sum_{x \sim p_{\theta^n}(x)} \left( W\Big(Q_i(\tilde{x}_i, \tilde{x}_{\mathrm{p}(i)}) - V_i(\tilde{x}_{\mathrm{p}(i)}) - \beta\Big) \log p_{\theta_i}(\tilde{x}_i \mid \tilde{x}_{\mathrm{p}(i)}) \right). \qquad (43)$$

There are a few minor differences from DADO. First, there's an entropy-maximizing term. Second, notice that the value function definitions are a little different from in DADO. Here we use $\log \sum \exp$ to define $V_i(\tilde{x}_{\mathrm{p}(i)})$, whereas DADO uses an expectation over the search distribution instead. Third, the $Q$-function has $V_i(\tilde{x}_{\mathrm{p}(i)})$ subtracted from it, whereas DADO's derivation doesn't. As mentioned above though, $V_i(\tilde{x}_{\mathrm{p}(i)})$'s role in the objective is that of a "baseline", and it can be changed out or dropped. In our implementation, we do subtract $V_i(\tilde{x}_{\mathrm{p}(i)})$—the expectation-based one, not $\log \sum \exp$—as a mean baseline function to reduce weight variance and obtain a more stable update (Sec. 4). Overall though, the result is quite similar to DADO (Sec. A.4), supporting the generality of the decomposition-aware distributional optimization framework, and suggesting that there are a variety of related objectives and algorithms one might use under this umbrella. In particular, one might adapt methods from RL. For example, Peng et al. (2019)'s derivation could be adapted to get a similar weighted maximum likelihood objective with the difference in value functions (also called the advantage) exponentiated instead. TRPO, PPO, or CbAS (Schulman et al., 2015; 2017; Brookes et al., 2019) might be adapted to regularize the search distribution toward some (identically factorized) prior, or the previous iteration's search distribution for stability. We emphasize that the core idea of DADO is to leverage message-passing (*i.e.*, value functions) on arbitrarily decomposed objectives to derive a distributional optimization procedure which updates search distribution factors separately, regardless of the specific objective and algorithm used.

**From classical value functions to distributional value functions.** Under this definition of value functions using $\log \sum \exp$ (Eq. 36), we can view $V_i(\tilde{x}_{\mathrm{p}(i)})$ as a soft maximum, which becomes a hard maximum when $Q_i$ are large, such that for $\tilde{x}_i^*$, $V_i(\tilde{x}_{\mathrm{p}(i)}) \approx Q_i(\tilde{x}_i^*, \tilde{x}_{\mathrm{p}(i)})$. The hard maximum case resembles classical message-passing, from which we began. When $Q_i$ are relatively small, information about multiple $\tilde{x}_i$ will pass through $\log \sum \exp$, thus propagating info for a distribution of descendant states in accordance with $p^*(\tilde{x}_i \mid \tilde{x}_{\mathrm{p}(i)}) \propto \exp Q_i(\tilde{x}_i, \tilde{x}_{\mathrm{p}(i)})/\beta$. Another way of seeing the transition from classical message-passing to distributional optimization is that for $\beta$ near 0, $p^*(\tilde{x}_i \mid \tilde{x}_{\mathrm{p}(i)})$ will concentrate all its mass on the $\arg\max$. But for a larger $\beta$, $p^*(\tilde{x}_i \mid \tilde{x}_{\mathrm{p}(i)})$ will concentrate on multiple designs proportional to their $Q$ values. As $\beta \to \infty$, $p^*(\tilde{x}_i \mid \tilde{x}_{\mathrm{p}(i)})$ becomes a uniform distribution.

### A.6 PROTEIN DATASET DETAILS

We investigated seven protein property datasets from Tareen et al. (2022) and Notin et al. (2023). These proteins span various lengths and all have more than single mutations away from wild-type. These datasets are:

- Adeno-associated virus 2 capsid protein (AAV2 capsid or AAV; $L = 28, D = 20$), with data from Bryant et al. (2021), which assayed 42,329 sequences for virus viability, including sequences with as many as 27 mutations from the wild-type. The full protein is of length 735. The dataset was accessed via Notin et al. (2023).

- Amyloid-beta (Amyloid; $L = 42, D = 21$, where the extra state is a stop codon), with data from Seuma et al. (2021), which assayed 16,066 sequences for aggregation with a nucleation score, 97% of which are double mutants. The dataset was accessed via Tareen et al. (2022).

- Yeast transcription factor Gcn4 ($L = 44, D = 20$), with data from Staller et al. (2018), which assayed 2,639 sequences for activity, including sequences with as many as 44 mutations from the wild-type. The full protein is of length 281. The dataset was accessed via Notin et al. (2023).

- TAR DNA-binding protein 43 (TDP-43; $L = 84, D = 21$, where the extra state is a stop codon), with data Bolognesi et al. (2019), which assayed 57,996 sequences for cell toxicity, 98% of which are double mutants. The dataset was accessed via Tareen et al. (2022).

- Protein G B1 domain (GB1; $L = 55, D = 20$), with data from Olson et al. (2014), who scanned all possible single and double mutants and assayed their 536,963 binding affinities to IgGFC, of which 99.8% are double mutants. The dataset was accessed via Tareen et al. (2022).

- ynzC, a small protein domain ($L = 39, D = 20$), with data from Tsuboyama et al. (2023), assayed for its folding stability on 2,301 variants, 68% of which are double mutants. The dataset was accessed via Notin et al. (2023).

- *Chlamydomonas reinhardtii*'s light-oxygen-voltage domain (CreiLOV or Phot; $L = 118, D = 20$), with data from Chen et al. (2023), which assayed 167,530 sequences for fluorescence, including sequences with as many as 15 mutations from the wild-type. The dataset was accessed via Notin et al. (2023).

Of these, the four shown in the main text were chosen according to two criteria. First, we wanted proteins for which the AlphaFold3-derived junction tree had relatively small nodes (*i.e.*, more decomposable proteins), discussed in more detail below. Second, we also prioritized proteins for which the datasets contained sequences many mutations away from the wild-type, as the resulting predictive models are more likely to be realistic in a larger area of the design space than datasets that only assay a concentrated ball of sequences. Based on these criteria, we decided to focus on AAV, Amyloid, Gcn4, and TDP-43 (Fig. 3). These four proteins' decomposition junction trees have nodes of cardinality five or fewer, whereas the junction trees for GB1 and ynzC have many nodes with cardinality over 10, and CreiLOV has nodes with cardinality greater than 20. Optimization performance on the remaining datasets is included in the appendix (Fig.A4).

We visualize all seven proteins' AlphaFold3-predicted structures and corresponding contact graphs and junction trees (Fig. A1). Panels 3–6 (the rightmost four columns) illustrate that AAV, Amyloid, Gcn4, and TDP-43 yield junction trees with smaller nodes and fewer positions overlapping in multiple junction tree nodes compared to GB1, ynzC, and CreiLOV. It would be interesting to reduce the contact threshold for the latter problems in order to obtain smaller nodes. As the possible efficiency gain from using a decomposition-aware method like DADO is determined by how sparse and tree-like this graph is, we expect DADO would yield more substantial performance improvements over standard EDAs in such a setting. Our studies of predictive model accuracy when varying the contact threshold (Figs. A6, A7, A9) suggest that one may be able to obtain almost-as-accurate predictive models under decomposition graphs with lower contact thresholds, but we did not further investigate optimization performance on these models.

## A.7 BASELINE METHOD DETAILS

We compare DADO with three baseline methods. The first is the naive, decomposition-unaware EDA, which we refer to as "EDA". Algorithmic details of this method are in Sec. 2 and Sec. A.2 also includes a derivation of the EDA algorithm. We take this as the starting point for the other baseline methods and now describe how the EDA is modified for each.

The second baseline we consider is the factorized distribution algorithm ("FDA") from Mühlenbein & Mahnig (1999). The only modification to the naive EDA is to replace its search distribution, ordinarily a joint distribution over all design variables, with a factorized search distribution. In all of our experiments, we give FDA the same search distribution factorization as DADO for fairness; as such, in our protein experiments, FDA's factorization is based on the AlphaFold3-based contact map. Comparing to FDA sheds light on how much of DADO's improved performance can be attributed to it using a factorized search distribution versus the contribution of using message-passing value functions for the factorized search distribution update because FDA only uses the former and not the latter, whereas DADO uses both.

The third baseline we consider is a proximal policy optimization version of the EDA, which we call "PPO". This baseline is inspired by Schulman et al. (2017). We only modify the EDA's update rule to match the PPO update rule. That is, instead of minimizing the naive EDA's loss, $L(\theta) = \mathbb{E}_{x^k \sim p_{\theta^n}(x)} \left[ f(x^k) \log p_\theta(x^k) \right]$, we instead minimize a proximal version of it,

$$L(\theta) = \mathbb{E}_{x^k \sim p_{\theta^n}(x)} \left[ \min \left\{ f(x^k) \frac{p_\theta(x^k)}{p_{\theta^n}(x)}, f(x^k) \, \text{clip}(\frac{p_\theta(x^k)}{p_{\theta^n}(x)}, 1 - \epsilon, 1 + \epsilon) \right\} \right],$$

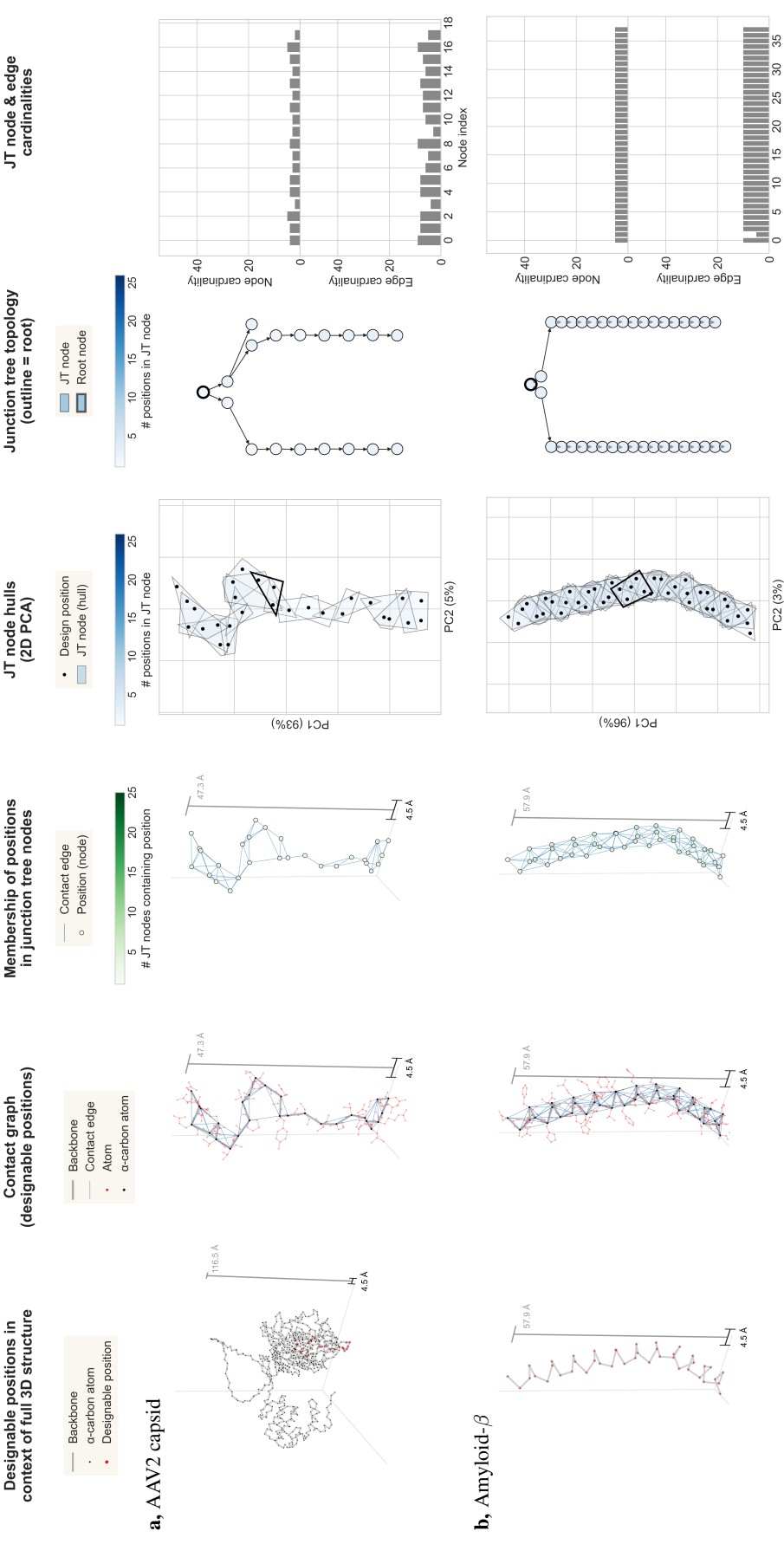

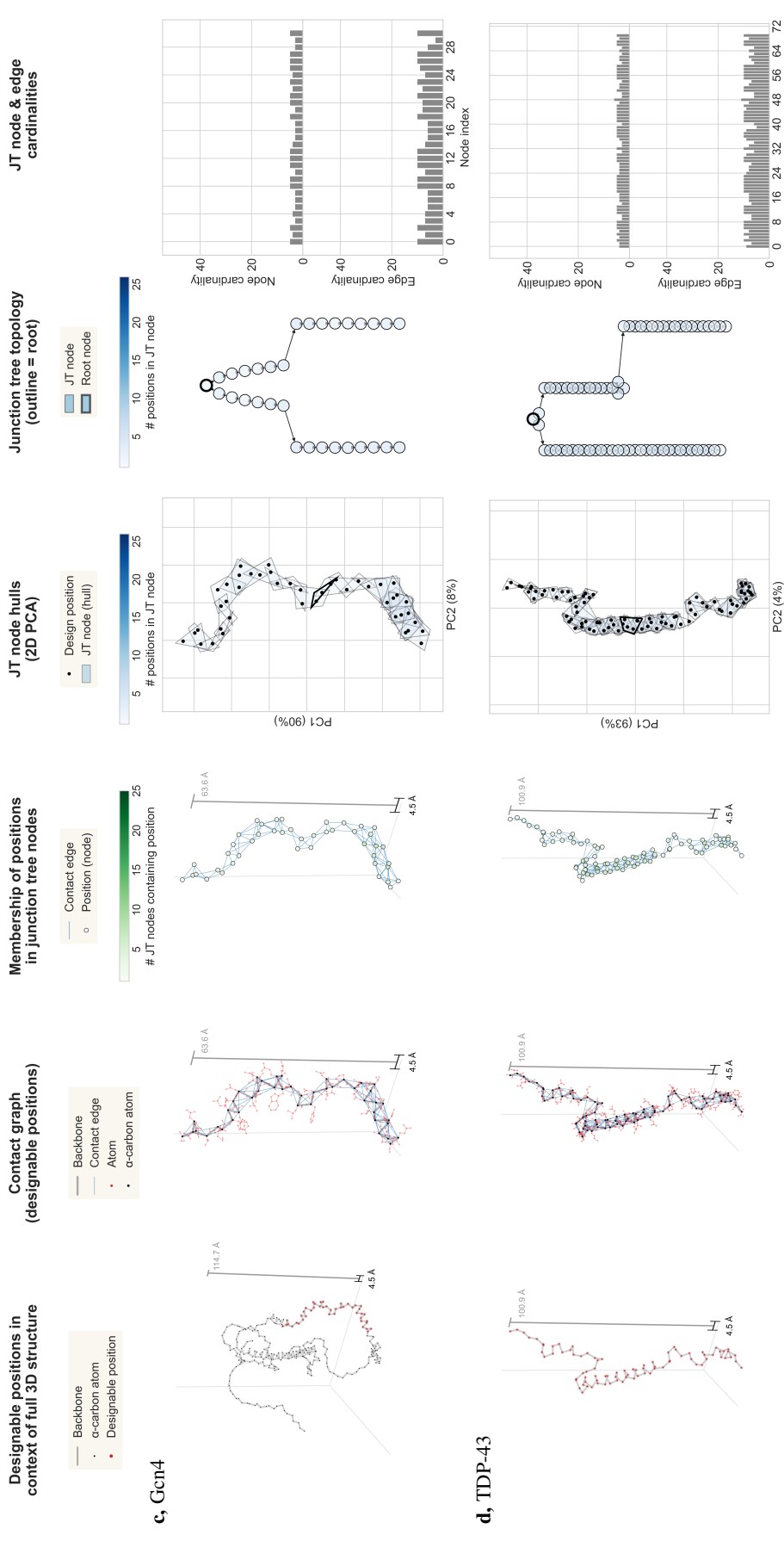

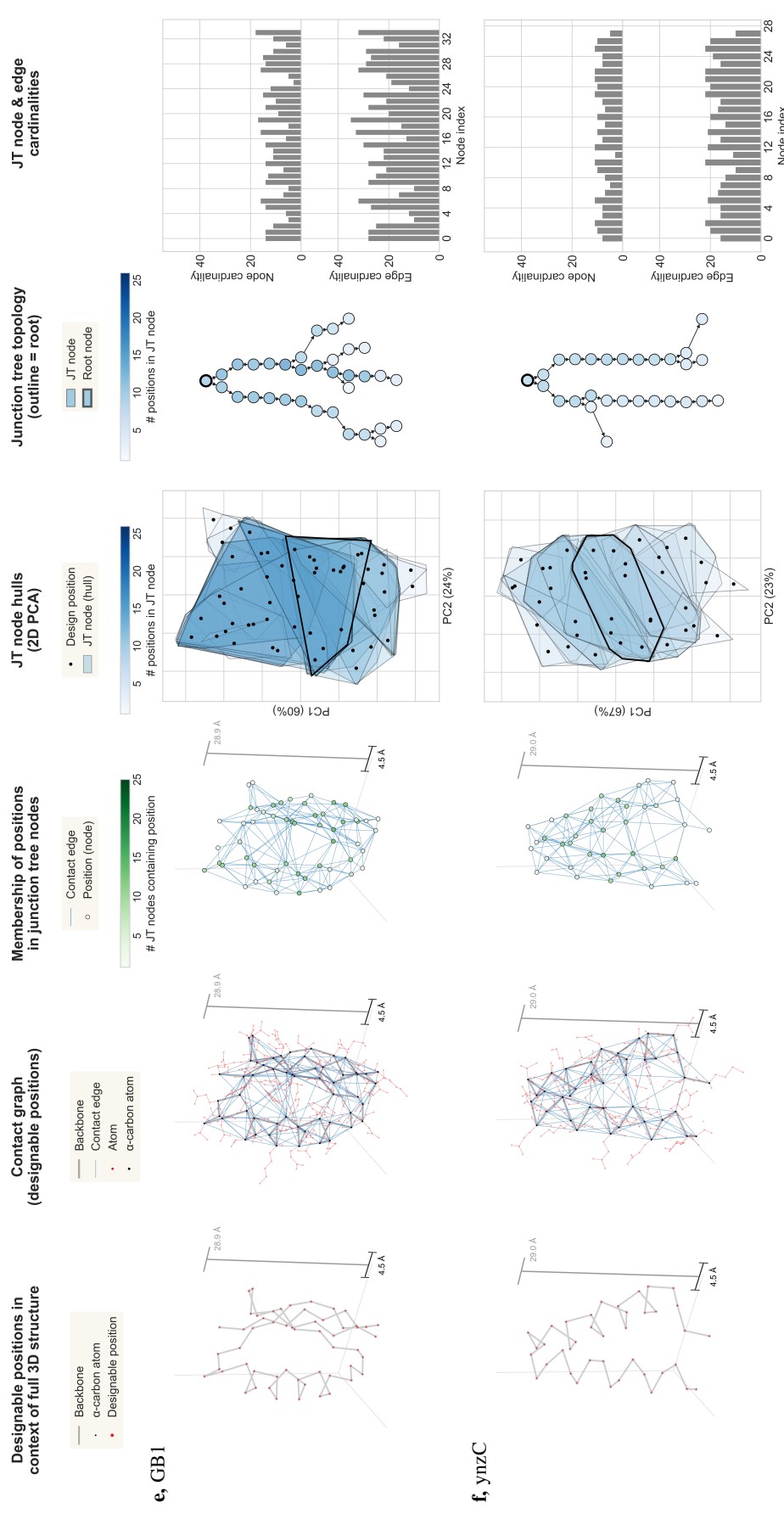

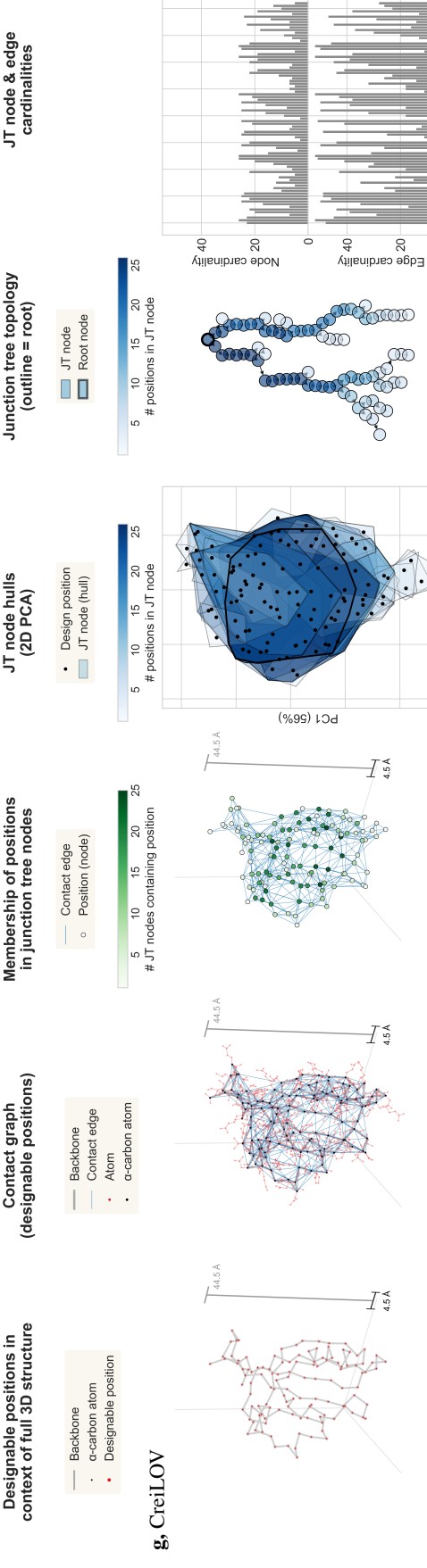

**Figure A1: 3D structure, contact graph, and junction tree decomposition for each protein.** For each of the seven protein datasets considered, **a,** AAV; **b,** Amyloid-$\beta$; **c,** Gcn4; **d,** TDP-43; **e,** GB1; **f,** ynzC; and **g,** CreiLOV, we show the AlphaFold3-predicted 3D structure and the contact graph and junction tree (JT) decomposition obtained by imposing a $4.5\ \text{Å}$ contact threshold. From left to right, each panel shows: **(1)** the full 3D backbone structure, with the $\alpha$-carbon atoms of designable positions highlighted in red; **(2)** the contact graph restricted to designable positions, where edges connect residue pairs with any atoms within $4.5\ \text{Å}$ of each other; **(3)** positions in the contact graph colored by how many JT nodes each participates in (global scale 1–25); **(4)** a 2D PCA projection of positions overlaid with convex hulls representing each JT node (hulls may encompass positions not contained as a result of dimensionality reduction), colored by cardinality (global scale 1–26) and with the root node marked by a thick black outline; **(5)** the directed JT topology, with nodes colored by cardinality (global scale 1–26) and the root node marked by a thick black outline; and **(6)** each node's cardinalities (top) and edge cardinalities—that is, of the intersection of the node's variables with its parent's variables—(bottom) with shared global scale (1–52). To examine high-resolution details (*e.g.,* 3D residue orientations), please view on a screen and zoom in.

where $\epsilon$ is a hyperparameter that we set to $0.2$ as suggested in the original PPO paper. PPO-style updates can be interpreted as stabilizing the search distribution update of the naive EDA by penalizing it from deviating too far from the previous iteration's search distribution. This approach could easily be added onto DADO and FDA; we did not explore this but expect it would be helpful for these decomposition-aware methods too.

## A.8 ADDITIONAL IMPLEMENTATION DETAILS

We record various implementation details for reproducibility (also see code released upon publication). We implemented all of our code using Jax, Flax, and Optax (DeepMind et al., 2020).

### A.8.1 SEARCH DISTRIBUTION

For the search distribution of DADO, we used MLP-based autoregressive neural networks for each position $l \in [1, \ldots, L]$—specifically, an MLP of size $[64, 64]$ that takes as input all conditioning variables (parents in the tree) and outputs $D$ logits for each step of autoregressive decoding. Our naive EDA implementation is the same, only assuming all variables are in one meta-variable—hence, a standard autoregressive model over all variables. The FDA baseline uses the same architecture as DADO, whereas our PPO baseline uses the same architecture as the naive EDA. For gradient descent to fit the search distribution, we used the AdamW optimizer (Loshchilov & Hutter, 2017) with default momentum parameters $\beta_1 = 0.9, \beta_2 = 0.999$.

### A.8.2 FULLY SYNTHETIC FUNCTIONS

We simulated one $f(x)$ for each of three sequence lengths, $L = \{25, 50, 200\}$, by first randomly sampling a junction tree topology, and then randomly specifying the component functions. Each node function, a $D$-vector, was sampled $f_i \in \mathbb{R}^D \sim \mathcal{N}(0, 0.01)$, where $\mathcal{N}$ denotes a Gaussian distribution. Each edge function, a $D \times D$ matrix, was sampled, $f_{i,j} \in \mathbb{R}^{D \times D} \sim \mathcal{N}(0, 0.0025)$. To ensure a reasonable degree of non-smoothness in $f(x)$, we further explicitly added what in biology is known as *reciprocal sign epistasis* (Starr & Thornton, 2016; Li et al., 2024). Specifically, for each edge function we, twice, randomly assigned one of the 20 alphabet letters to each node, i) $x_i := A, x_j := B$ and ii) $x_i := C, x_j := D$. Next we sampled an effect size, $\lambda \sim \mathcal{N}(0, 4)$. Finally, we set $f_{i,j}(x_i = A, x_j = B) = 0$ and $f_{i,j}(x_i = C, x_j = B) = \lambda$, and also half of the time, $f_{i,j}(x_i = A, x_j = D) = 0$ and $f_{i,j}(x_i = A, x_j = D) = \lambda$.

### A.8.3 DECOMPOSED PROTEIN PROPERTY PREDICTIVE MODELS

For each dataset, we used AlphaFold3-predicted structures (Abramson et al., 2024) on the wild-type sequence to obtain a 3D structure, from which we constructed a contact graph by thresholding the distance between pairs of residues with threshold $t$. Following Brookes et al. (2022); Romero et al. (2013); Voigt et al. (2002), we use a threshold of $t = 4.5$Å. We interpret this contact map as a graph adjacency matrix, from which we algorithmically construct a junction tree (Lauritzen & Spiegelhalter, 1988). This defines the topology needed for DADO, but we must also fit the component functions on the protein assay-labeled data. To do so, we employ an MLP-based predictive model that strictly enforces the decomposition defined by the junction tree. In particular, we use an MLP that takes as input the sequence, and, critically, also a bit-vector specifiying active variables. One function evaluation requires calling this MLP for every node and edge function and summing. We used 5-fold cross validation to sweep through hidden layers of size ($[16, 16]$, $[128, 16]$, or $[128, 128, 16]$), learning rate ($0.001$ or $0.0001$), and number of training iterations ($5,000$ or $50,000$), to choose hyperparameters with the lowest cross-validation mean-squared error. Finally, we trained the model with those hyper-parameters using all data available.

## A.9 ADDITIONAL EXPERIMENTAL RESULTS

Herein, we include additional experimental results on synthetic landscapes for completeness—primarily studying the relationship between performance and number of design variables ($L$), and larger alphabet sizes ($D$). For protein landscapes, we also perform a case study on one protein dataset in order to study the effect of the distance threshold, $t$, on decomposability, predictive accuracy, and downstream optimization performance.

### A.9.1 SYNTHETIC FUNCTIONS: L=400

We wanted to compare DADO and a naive EDA on even larger design spaces, but this can take quite a while to run when considering the hyperparameter sweep and replicates (Sec. A.9.7). For expediency, we chose the same hyperparameters (temperature and learning rate) as were used by each method for $L = 200$ (Fig. 2). We then ran 20 replicates of each method.

At $L = 400$, the trend we observed in Fig. 2 continues—as $L$ grows, the gap between DADO and the EDA grows increasingly larger (Fig. A2), as expected, because the full design space grows at a faster rate than the decomposed design space that DADO operates in.

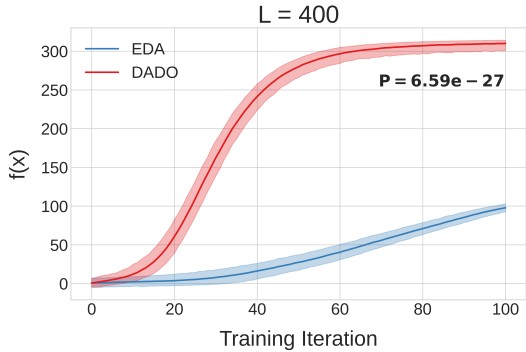

**Figure A2: Comparison of a naive EDA to DADO on a synthetic problem with** $L = 400$. We created a random function, $f(x)$, with a randomly chosen junction tree decomposition with maximum node size of one, and randomly chosen parameters. We used alphabet size $D = 20$ and sequence length $L = 400$. Each of the two methods drew $K = 100$ samples per iteration. For each iteration, we show the mean (solid line) and 95% confidence interval (shaded envelope) of the 100 samples evaluated on $f(x)$, averaged across results from 20 random seeds. P-value is from a two-sided paired t-test that the mean at the final iteration is different between methods, over the 20 seeds.

### A.9.2 SYNTHETIC FUNCTIONS: INCREASING D

For all of our synthetic experiments, we used an alphabet size of $D = 20$, which reflects the typical alphabet for protein design problems. Design problems in other scientific domains might have larger alphabets so we also considered $D = 50$ and $D = 100$ here, keeping $L$ fixed at 100.

We observe that DADO finds designs with higher $f(x)$ than the naive EDA across all three alphabet sizes (Fig. A3). As one would expect, as $D$ grows and the design space grows combinatorially larger, it becomes increasingly difficult for both methods to optimize $f(x)$. In general, such problems require using a larger sampling budget ($K$) and/or more iterations. Although the performance gap between DADO and EDA seems to shrink as $D$ grows, the EDA has converged to a suboptimal region of the design space, whereas DADO still has a positive slope and a lot of sampling diversity. This suggests that were we to run more iterations, DADO would continue to improve, but the EDA would not.

### A.9.3 COLLECTED PROTEIN EXPERIMENTS

Here, we collect the four proteins shown in the main text (Fig. 3) alongside three other proteins we tested (details of why those four were chosen in Sec. A.6). First, we plot them with $-\log(c - f(x))$ on the y-axis, for clarity when $f(x)$ is high (Fig. A4). For comparison, we then show the same experimental results with just $f(x)$ on the y-axis (Fig. A5). Whereas AAV, Amyloid, Gcn4, and TDP-43 have decomposition junction trees with nodes of cardinality five or less, the junction trees for GB1 and ynzC have many nodes with cardinality over 10, and CreiLOV has nodes with cardinality greater than 20. For GB1 and ynzC, DADO still outperforms the decomposition-unaware baselines, but not FDA, which also operates in the decomposed design space. We hypothesize that FDA performs well relative to DADO in these cases because as the junction tree nodes grow larger, the computed value functions become higher variance estimates, such that it can be better to weight the search distribution by $f(x)$ directly. CreiLOV has even larger nodes, potentially amplifying

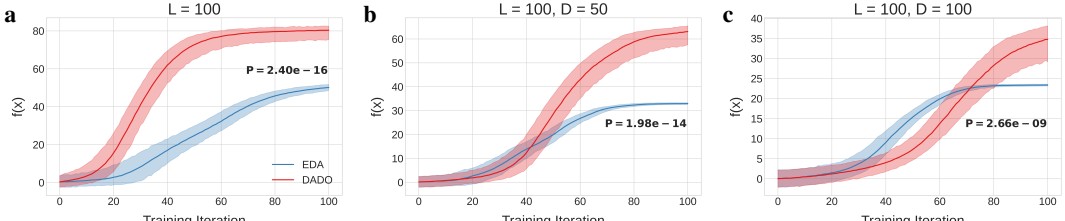

**Figure A3: Comparison of a naive EDA to DADO on synthetic problems with large alphabets.** We created three random functions, $f(x)$, each with the same randomly chosen junction tree decomposition with maximum node size of one, and randomly chosen parameters. Each experiment used sequence length $L = 100$ and alphabet size **a,** $D = 20$, **b,** $D = 50$, and **c,** $D = 100$. Each of the two methods drew $K = 100$ samples per iteration. For each iteration, we show the mean (solid line) and 95% confidence interval (shaded envelope) of the 100 samples evaluated on $f(x)$, averaged across results from 20 random seeds. P-values are from a two-sided paired t-test that the mean at the final iteration is different between methods, over the 20 seeds.

this effect. For visualization and comparison of junction trees across proteins, see Fig. A1. It would be interesting to test this hypothesis by implementing further variance-reduction techniques for DADO's value functions.

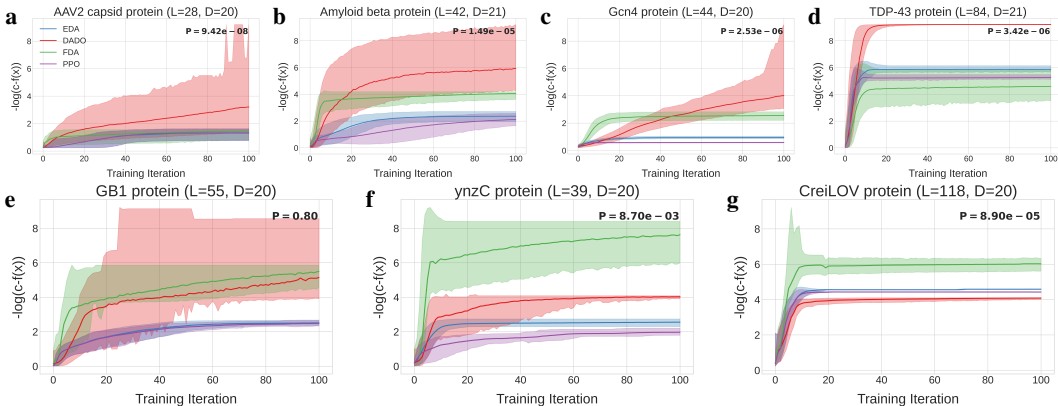

**Figure A4: Optimization performance on protein problems, plotted on a negative log scale.** For each of seven proteins of varying length, **a,** AAV (also Fig. 3a), **b,** Amyloid (also Fig. 3b), **c,** Gcn4 (also Fig. 3c), **d,** TDP-43 (also Fig. 3d), **e,** GB1, **f,** ynzC, and **g,** CreiLOV, we fit a neural network property function, $f(x)$, adhering to a junction tree decomposition derived from the protein's 3D structure, and then used standard EDA and DADO to optimize them. Each approach drew $K = 1000$ samples per EDA iteration. For each iteration, we show the mean (solid line) and 95% confidence interval (shaded envelope) of the 1000 samples evaluated on $-\log(c - f(x))$, averaged across results from 20 random seeds. We plot this quantity to make clear the differences between methods when $f(x)$ is large; $c$ is the largest $f(x)$ on a given plot, plus a small constant for numerical stability. P-values are from a two-sided paired t-test that the mean at the final iteration is different between methods, over the 20 seeds.

### A.9.4 INVESTIGATION OF PROTEIN PROPERTY PREDICTIVE MODEL DECOMPOSABILITY

We also sought to investigate how changing the threshold that determines the complexity of the junction tree would affect both predictive performance and optimization performance. Specifically, we varied the distance threshold, $t$, for which pairs of residues were considered contacting to explore the tradeoff between accuracy of the model and decomposability. The lower the value of $t$, the stricter the decomposition (the smaller the cardinality of the largest meta-variable); thus lower $t$ should give DADO a larger advantage over the standard EDA, but may be overly restrictive and yield a worse predictive model. We decided to do a case study of GB1 for $K = 100$ samples at each iteration because in this setting, using the default threshold of $t = 4.5$Å leads to comparable performance between DADO and the EDA. We wanted to see if decomposing the model further would result in a function that's easier for DADO to optimize without sacrificing accuracy. We find that the largest $t$ provides the highest holdout predictive accuracy, which is expected because the resulting decomposed model isn't restricted. However, decreasing this distance down to $t = 2.75$Å allows

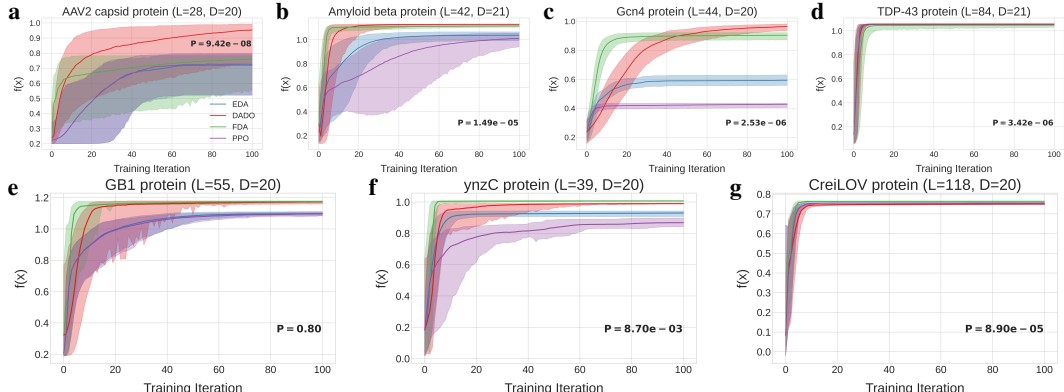

**Figure A5: Optimization performance on protein problems, plotted on a standard scale.** For each of seven proteins of varying length, **a,** AAV (also Fig. 3a), **b,** Amyloid (also Fig. 3b), **c,** Gcn4 (also Fig. 3c), **d,** TDP-43 (also Fig. 3d), **e,** GB1, **f,** ynzC, and **g,** CreiLOV, we fit a neural network property function, $f(x)$, adhering to a junction tree decomposition derived from the protein's 3D structure, and then used standard EDA and DADO to optimize them. Each approach drew $K = 1000$ samples per EDA iteration. For each iteration, we show the mean (solid line) and 95% confidence interval (shaded envelope) of the 1000 samples evaluated on $f(x)$, averaged across results from 20 random seeds. P-values are from a two-sided paired t-test that the mean at the final iteration is different between methods, over the 20 seeds.

most of the predictive signal to remain while substantially reducing the complexity of the junction tree (Fig. A6a), enabling DADO to consistently converge on designs with high $f(x)$ within only 10 iterations (Fig. A6b), whereas for $t = 4.5\text{Å}$ and $t = 9\text{Å}$, DADO's mean does not clearly converge even after 100 iterations, and its distribution remains dispersed as evidenced by wide shaded envelopes (Fig. A6c,d). Additionally, DADO definitively outperforms the decomposition-unaware EDA when $t = 2.75\text{Å}$ (Fig. A6b). When the predictive model is less decomposed, DADO is not as distinguishable from the EDA (Fig. A6c,d).

### A.9.5 DECOMPOSITION GRAPH ROBUSTNESS

How crucial is it to guess the decomposition perfectly *a priori*? One way to investigate this is to mutate the decomposition graph and observe how the resulting decomposed predictive models' predictive accuracies change. In our first set of experiments, we varied the contact threshold ($t$) used on the AlphaFold 3D structure to determine connectivity; lowering $t$ will gradually remove more distant contacts, whereas increasing $t$ will gradually add more distant contacts. Given different decomposition graphs, we then trained a decomposed predictive model for each. For our second set of experiments, we performed random mutations to the $t = 4.5\text{Å}$ decomposition graph (used in Sec. 4) to study how robust prediction is to missing / extra edges. In particular, for each of $N \in [-50, -10, -5, -1, 1, 5, 10, 50]$, we randomly sampled $N$ edges to remove/add. We implemented a check to ensure that the graph doesn't become disconnected, so for some experiments, we cut off $N$ at the largest number of edges that could be removed resulting in a chain graph. We repeated this procedure 10 times.

In both of our experiments, we considered holdout accuracy as measured by the Pearson correlation coefficient between assay labels and model predictions. All decomposed models for each protein used the same hyperparameters that were chosen via 5-fold cross validation on the full dataset for the base $t = 4.5\text{Å}$ decomposed model, for expediency. We also compared to a neural network without any decomposition (*i.e.*, an all-edges model), denoted "Naive NN" in our plots. For this model we performed an additional 5-fold cross validation on the full dataset for each protein.

We observe that the holdout accuracy of the non-decomposed predictive model and the decomposed models of varying $t$ all tend to fall within a relatively small range (Fig. A7), suggesting that using our decomposed predictive model does not constitute a substantial sacrifice compared to a full (naive) model, and that our decomposition is somewhat robust to changing the contact threshold. When we randomly mutate the decomposition graph, we find that holdout accuracy is generally concentrated within an even smaller range, with some exceptions when many edges are added or removed (Fig. A8). Interestingly, for Gcn4, the decomposed predictive models outperform the naive model,

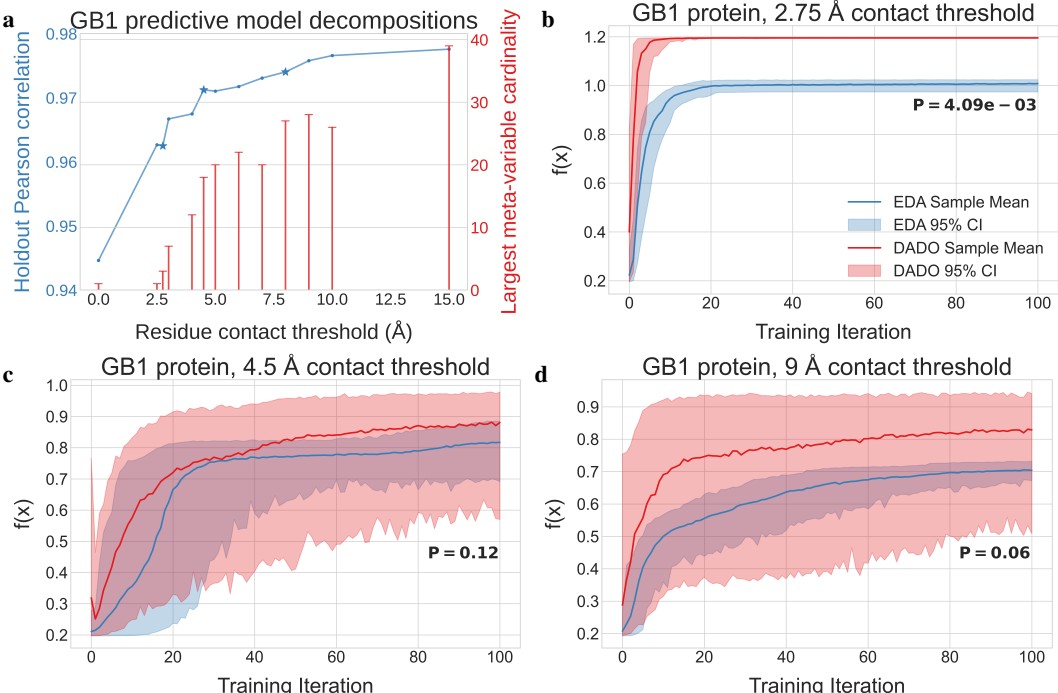

**Figure A6: Investigation of GB1 predictive model decomposability. a,** We varied the distance threshold, $t$, for which pairs of residues in protein GB1's 3D structure are considered neighbors in the junction tree, using several values in $t \in [0\text{Å}, 15\text{Å}]$, so as to explore the tradeoff between accuracy of the model and decomposability. The lower the value of $t$, the stricter the decomposition (the smaller the cardinality of the largest meta-variable); thus lower $t$ should give DADO a larger relative advantage over the standard EDA, but may yield a worse predictive model from imposing a more restrictive functional structure. The Pearson correlation on a 10% holdout set remains quite good down to and including $t = 2.75\text{Å}$, a point at which DADO provides a statistically significant win over a standard EDA as seen in panel b. Blue stars denote values of $t$ corresponding to the experiments in panels **b,** $t = 2.75\text{Å}$, **c,** $t = 4.5\text{Å}$, and **d,** $t = 9\text{Å}$. Each method drew 100 samples per iteration. For each iteration, we show the mean (solid line) and 95% confidence interval (shaded envelope) of the 100 samples evaluated on $f(x)$, averaged across results from 20 random seeds that dictated initialization of the search distribution. P-values are from two-sided paired t-tests that AUC of the per-iteration mean is different between methods, using the 20 mean curves.

though it's worth noting that this prediction task is especially hard due to the relatively uniform dispersion of sequences throughout the design space, and the small size of the dataset. Both models fit the training data well (not shown here).

### A.9.6 DECOMPOSITION GRAPH ROBUSTNESS: BOTTOM 50% DATA

Herein, we repeated the same analyses as in Sec. A.9.5, except using the bottom half of the training set (*i.e.*, the datapoints with the lowest assay labels). We used the same exact holdout sets as in Sec. A.9.5, such that decomposed predictive models are tested against sequences with both high and low assay labels. We used the same hyperparameters for all of the predictive models before, which were cross-validated on the full dataset.

Overall, we observe that compared to Fig. A7 and Fig. A8, holdout accuracy as measured by the Pearson correlation coefficient between assay labels and model predictions is lower for all models and all proteins (Fig. A9, Fig. A10). Interestingly, for several proteins, the more decomposed predictive models (lower $t$) performed better than the less decomposed models (higher $t$), as well as the full (naive) model (Fig. A9). This makes sense, as more complex models are more prone to overfitting when the training data is shifted from the holdout distribution, and as the training dataset gets smaller. This trend is also reflected in the random graph mutation experiments (Fig. A10). Generally, the same trends observed in Sec. A.9.5 hold here, such as predictive accuracy being relatively robust to changing $t$ and to randomly mutating the graph (up to a point).

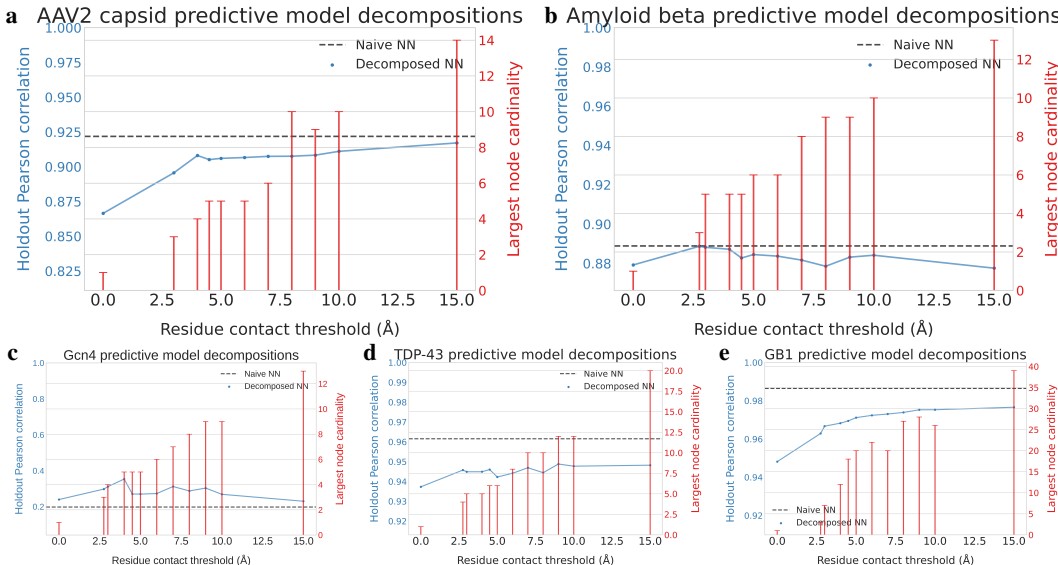

**Figure A7: Investigation of predictive model decomposability by contact threshold.** We varied the contact threshold, $t$, for which pairs of residues in each protein's 3D structure are considered neighbors in the decomposition graph, using several values in $t \in [0\text{Å}, 15\text{Å}]$, so as to explore the tradeoff between accuracy of the model and decomposability. We studied **a,** AAV, **b,** Amyloid, **c,** Gcn4, **d,** TDP-43, and **e,** GB1.

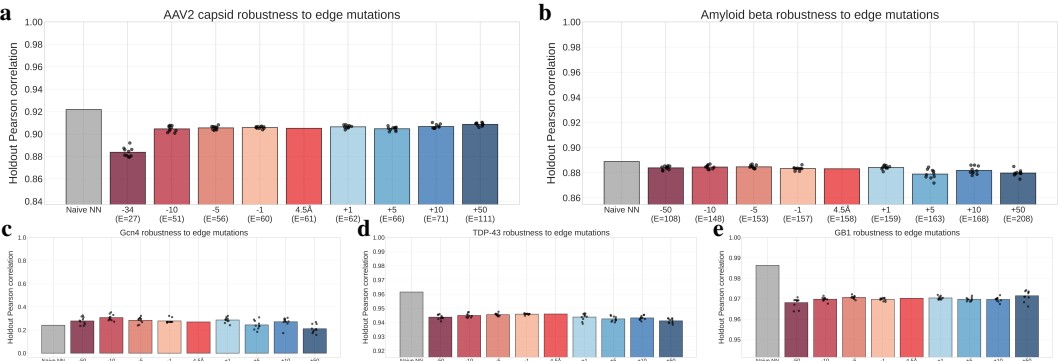

**Figure A8: Investigation of predictive model decomposability by random graph mutation.** For each of **a,** AAV, **b,** Amyloid, **c,** Gcn4, **d,** TDP-43, and **e,** GB1, we randomly added or removed edges from the $t = 4.5\text{Å}$ decomposition graph. In particular, we randomly sampled $N \in [-50, -10, -5, -1, 1, 5, 10, 50]$ edges to remove/add. For certain experiments we limit $N$ to avoid creating disconnected graphs. We repeated this procedure 10 times.

### A.9.7 RUNTIME ANALYSIS

We measure the wall-clock time it takes to run a single distributional optimization algorithm for both DADO and the naive EDA (Table 1). We fix both to run for 100 iterations, drawing 100 samples at each, using the same architectures as described in Sec. A.8.1, and on a single GPU. We report times just to give a rough sense of runtime and scaling; one could definitely further optimize our code.

A few trends stand out. First, the EDA and DADO take roughly the same amount of time to run, with DADO being faster sometimes. This speed-up may come from using a factorized search distribution, which requires only conditioning on parent variables as opposed to all preceding variables (autoregressive). In general, problems with larger $L$ take longer to run, and this scaling is worse than linear for our implementation. We also notice that when increasing $D$ and holding $L$ fixed at 100, the runtime is similar. A single run of either DADO or EDA does not generally require the full memory of a 16GB GPU, so in practice, we're able to parallelize runs within GPUs and across multiple GPUs in order to perform our hyperparameter sweep and random seed replicates efficiently.

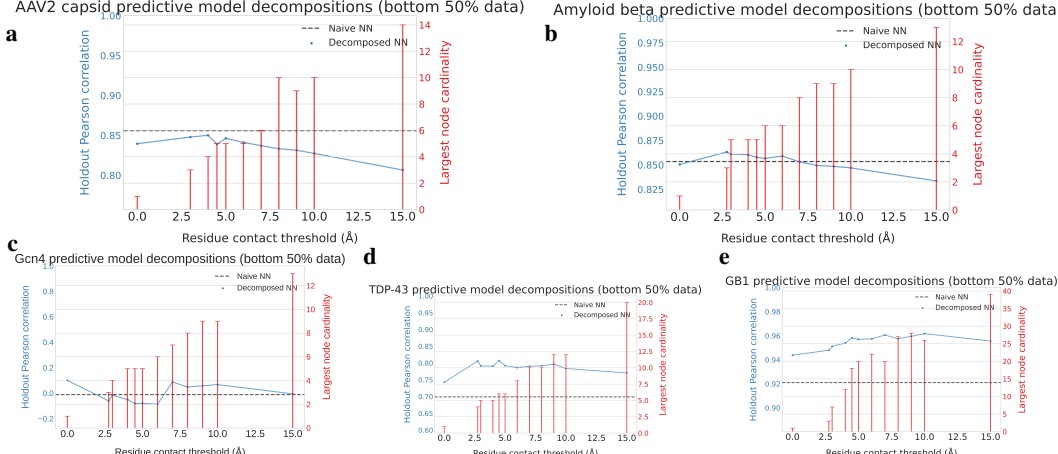

**Figure A9: Investigation of predictive model decomposability by contact threshold (bottom 50% data).** We varied the contact threshold, $t$, for which pairs of residues in each protein's 3D structure are considered neighbors in the decomposition graph, using several values in $t \in [0\text{Å}, 15\text{Å}]$, so as to explore the tradeoff between accuracy of the model and decomposability. We studied **a,** AAV, **b,** Amyloid, **c,** Gcn4, **d,** TDP-43, and **e,** GB1. We used the same holdout set as in Fig. A7, but only trained on the bottom 50% of data by assay label.

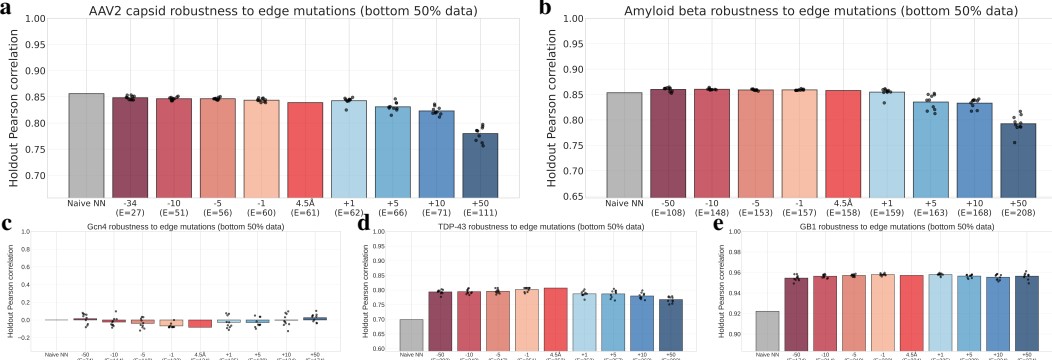

**Figure A10: Investigation of predictive model decomposability by random graph mutation (bottom 50% data).** For each of **a,** AAV, **b,** Amyloid, **c,** Gcn4, **d,** TDP-43, and **e,** GB1, we randomly added or removed edges from the $t = 4.5\text{Å}$ decomposition graph. In particular, we randomly sampled $N \in [-50, -10, -5, -1, 1, 5, 10, 50]$ edges to remove/add. For certain experiments we limit $N$ to avoid creating disconnected graphs. We repeated this procedure 10 times. We used the same holdout set as in Fig. A8, but only trained on the bottom 50% of data by assay label.

**Table 1: Runtime comparison of EDA and DADO, varying $L$ and $D$.**

| Problem | Length (L) | Alphabet size (D) | EDA runtime | DADO runtime |
|---|---|---|---|---|
| Synthetic tree | 25 | 20 | 2.8 min. | 2.9 min. |
| Synthetic tree | 50 | 20 | 6.3 min. | 7.0 min. |
| Synthetic tree | 100 | 20 | 16.3 min. | 16.1 min. |
| Synthetic tree | 200 | 20 | 42.7 min. | 36.2 min. |
| Synthetic tree | 400 | 20 | 139.5 min. | 114.2 min. |
| Synthetic tree | 100 | 50 | 14.0 min. | 16.0 min. |
| Synthetic tree | 100 | 100 | 14.5 min. | 18.3 min. |
| AAV2 capsid protein | 28 | 20 | 2.1 min. | 3.4 min. |
| TDP-43 protein | 84 | 21 | 6.7 min. | 10.5 min. |

