# OpenReview forum: "Leveraging Discrete Function Decomposability for Scientific Design"
_ICLR.cc/2026/Conference — ICLR 2026 Poster_

### Official Review · Reviewer_dP4b · 2025-10-21

**Soundness:** 2
**Presentation:** 2
**Contribution:** 3
**Rating:** 4
**Confidence:** 3

**Summary:**

This paper introduces Decomposition-Aware Distributional Optimization (DADO) as a new algorithm for optimizing discrete functions that exhibit decomposable structures. The method learns a decomposed functional form of an objective and then incorporates a factorized search and message-passing schema to coordinate updates. The method is evaluated against Estimation of Distribution Algorithm (EDA) on both synthetic and real protein design tasks.

**Strengths:**

- To the best of my knowledge, the core idea of applying message passing for distributional optimization is novel and an interesting approach to the problem of scientific design.
- I appreciate the statistical tests reported in the manuscript.

**Weaknesses:**

At a high level, my main concerns/questions stem from the learned decomposition of the objective function. While there is evidence that DADO works if a good function decomposition exists, it is not clear that (1) such a good function decomposition always exists; and (2) that the method outperforms black box optimization methods that do not even take advantage of a function decomposition.

1. In the paragraph starting from line 100, the manuscript describes a method to "guess" the functional form of the predictive model and then fitting a model that enforces the guessed decomposition to the available data. However, this might result in a potential distribution shift between the fitted model and the true objective, which may or may not adhere to the "guessed" functional form. This is of particular concern in situations where the "high scoring" candidate designs are rare in the training data and might obey a different underlying functional form compared to other designs. It would be nice to better understand how much this possible distribution shift is actually a concern.
2. Somewhat related to the above comment, but it would be good to ablate the size of the training dataset used to learn the MLP-based function decomposition, instead of using all the data available (line 451). In particular, I feel it would be important to investigate the performance of DADO if only the bottom $x$th percentile of designs were used to learn the surrogate function.
3. The manuscript seems to specifically focus on applications of DADO for protein design (in addition to the synthetic function testing). However, it seems from the title, abstract, introduction, and other parts of the text that DADO is proposed as a general method for many different possible discrete design problems in AI4Science. In this light, it would be good to include additional evaluations across different domains (e.g., circuit, molecule, and material design, for instance) to better illustrate how DADO actually generalizes across different scientific domains.
4. It seems like a number of baselines are missing from experimental evaluation - in particular, how does DADO compare with methods that do not involve learning a junction tree decomposition at all? This would include any black-box optimization method - PPO, BO, FDA, MCTS (some of which are non-distributional but can still be evaluated using the distributional optimization framework). This would help better clarify the added benefit of even learning a functional decomposition in the first place.
5. It would be good to include results on ablating $K$ in step 1 (line 154).

**Questions:**

6. Is it ever possible for the functional form to enforce the wrong prior over the input space (for example, if the training data is messy, noisy, affected by a confounding variable, or otherwise unreliable)? If so, what happens to the performance of DADO in these situations?

---

> ### Author Response · Authors · 2025-11-27
> **Response to reviewer dP4b**
>
> > It seems like a number of baselines are missing from experimental evaluation - in particular, how does DADO compare with methods that do not involve learning a junction tree decomposition at all? This would include any black-box optimization method - PPO, BO, FDA, MCTS
>
> We thank the reviewer for the specific baseline suggestions. We added FDA and PPO, as these methods most cleanly integrate with the distributional optimization setup. Across both synthetic and protein tasks, DADO continues to outperform the decomposition-unaware baselines (EDA and PPO), as well as a partially-decomposition-aware baseline (FDA), so our conclusion about DADO’s improvement remains unchanged.
>
> > [Methods of guessing a decomposition] might result in a potential distribution shift between the fitted model and the true objective, which may or may not adhere to the "guessed" functional form. This is of particular concern in situations where the "high scoring" candidate designs are rare in the training data and might obey a different underlying functional form compared to other designs. It would be nice to better understand how much this possible distribution shift is actually a concern….​​Is it ever possible for the functional form to enforce the wrong prior over the input space (for example, if the training data is messy, noisy, affected by a confounding variable, or otherwise unreliable)? If so, what happens to the performance of DADO in these situations?
>
> It’s certainly possible to enforce a poor choice of decomposed functional form. Empirically, however, our decomposed predictors achieve holdout accuracy comparable to all-edges (dense) predictors across datasets (e.g., for AAV, Pearson ~0.92 vs. ~0.91; Fig. A6a), indicating that decomposition is not a limiting factor for the protein tasks considered. When we randomly add/delete edges (Fig. A7) or vary the contact threshold (Fig. A6), holdout accuracy remains stable, suggesting robustness to moderate decomposition error.
>
> Prior work (e.g., lines 40-45) extensively studies the effects of distribution shift when performing optimization with imperfect predictors. Since our decomposed predictors do not suffer from distribution shift any more than dense predictors do, we rely on this literature rather than proposing decomposition-specific solutions to these broad, well-known issues.
>
> > it would be good to ablate the size of the training dataset used to learn the MLP-based function decomposition, instead of using all the data available (line 451). In particular, I feel it would be important to investigate the performance of DADO if only the bottom th percentile of designs were used to learn the surrogate function.
>
> At the reviewer’s suggestion, we also evaluated predictive accuracy when only trained on the bottom 50% of designs (Fig. A8, A9). Relative to using all available data (Figs. A6, A7), we observe a modest decrease in holdout accuracy. Notably, in this low-$f(x)$ training data regime, the decomposed predictors are often more accurate out of distribution than dense models, consistent with more complex models overfitting.
>
> > it is not clear that (1) such a good function decomposition always exists
>
> We agree that good decompositions do not always exist, and obtaining them can be nontrivial even when they do. Our claim is narrower: in domains where approximate decompositions are available, DADO can improve optimization efficiency. Our protein experiments illustrate one such domain, and proteins admit multiple plausible sources of decomposition (e.g., AlphaFold, functional annotation, alignment, clustering, as pointed out by reviewer 6Cz8). Broader discovery of decompositions remains an active research direction (lines 530-533).
>
> > it seems from the title, abstract, introduction, and other parts of the text that DADO is proposed as a general method for many different possible discrete design problems in AI4Science. In this light, it would be good to include additional evaluations across different domains (e.g., circuit, molecule, and material design, for instance) to better illustrate how DADO actually generalizes across different scientific domains.
>
> As stated in the introduction (lines 45-46), we present protein design as a proof of concept, even though the method itself is general. Extending DADO to other domains—such as circuits, molecules, or materials—calls for domain-specific strategies for constructing or learning decompositions, which is beyond the scope of this paper, or data-driven decomposition learning (lines 530-533).

---

### Official Review · Reviewer_6Cz8 · 2025-10-28

**Soundness:** 3
**Presentation:** 3
**Contribution:** 3
**Rating:** 4
**Confidence:** 3

**Summary:**

The paper addresses a critical bottleneck in AI-driven scientific design—scaling distributional optimization to high-dimensional discrete spaces (e.g., proteins, circuits)—by exploiting function decomposability. Its contributions are well-aligned with the needs of both machine learning (ML) for optimization and domain sciences (e.g., protein engineering), making it a valuable addition to the field.

**Strengths:**

1. Unlike standard Estimation of Distribution Algorithms (EDAs) or reinforcement learning (RL) policy optimization, DADO explicitly leverages the decomposability of objective functions (via junction trees) to avoid optimizing over intractable full combinatorial spaces. This is a departure from "black-box" optimization approaches that treat the objective as monolithic.
 2. It generalizes classical max-plus message passing (used for exact global optimization) to distributional optimization, replacing hard maximization with soft, sample-based expectations. This enables DADO to balance exploration (via a factorized generative model) and exploitation (via coordinated message passing)—a key innovation not seen in prior work like Factorized Distribution Algorithms (FDAs) or chain-structured RL policies.
3. The paper also explores a practical tradeoff: tuning decomposability (via residue distance thresholds in proteins) to balance predictive model accuracy and optimization efficiency. This empirical insight bridges theoretical algorithm design and real-world scientific constraints.

**Weaknesses:**

1. The paper relies heavily on structure-based decomposability (AlphaFold3 contact graphs) for proteins but does not explore other practical sources of decomposability: For example, in protein design, decomposability could also be derived from sequence homology (conserved vs. variable regions) or functional annotations (binding sites vs. structural loops). Similarly, in circuit design, decomposability might come from modular components.
2. The paper empirically shows that "loose" decomposability (e.g., t=2.75 Å for GB1) retains predictive accuracy, but lacks theoretical bounds on how much decomposition error DADO can tolerate.
3. The paper uses D=20 (amino acids) for all experiments but does not address scalability to larger discrete alphabets (e.g., D=100 for small molecules or circuit components)

**Questions:**

1. For domains without 3D structural data (e.g., novel peptides or synthetic materials), what alternative methods would you recommend to derive decomposability for DADO? For example, could unsupervised learning (e.g., clustering design variables by co-occurrence) be used to
2. If you intentionally introduce errors into the junction tree (e.g., remove 10% of true residue contacts for TDP43), how much does DADO’s performance degrade relative to using the correct tree? Are there any heuristics (e.g., adding "redundant" edges to the junction tree) to mitigate this error?
3. For D=50 (e.g., small molecules with 50 building blocks), would DADO’s current MLP-based search distribution require prohibitive compute? If so, what modifications would you propose to scale DADO to larger alphabets?

---

> ### Author Response · Authors · 2025-11-27
> **Response to reviewer 6Cz8**
>
> > does not address scalability to larger discrete alphabets (e.g., D=100 for small molecules or circuit components)...For D=50 (e.g., small molecules with 50 building blocks), would DADO’s current MLP-based search distribution require prohibitive compute?
>
> DADO scales to larger alphabet sizes ($D$) and sequence lengths ($L$). We added results for $D=50,100$ (Fig. A2), $L=200$ (Fig. 2), and $L=400$ (Fig. A1). In all cases, DADO outperforms a decomposition-unaware EDA. Runtime is not a bottleneck for any of these settings (table in Sec. A.8.7).
>
> > If you intentionally introduce errors into the junction tree (e.g., remove 10% of true residue contacts for TDP43), how much does DADO’s performance degrade relative to using the correct tree? Are there any heuristics (e.g., adding "redundant" edges to the junction tree) to mitigate this error?
>
> When we randomly add/delete edges (producing slightly incorrect decompositions; Fig. A7) or vary the contact threshold (producing incomplete decompositions; Fig. A6), holdout accuracy remains stable, indicating tolerance to moderate decomposition error.
>
> Adding edges to reduce decomposition errors is a sensible strategy, but trades off with model complexity: overly dense graphs can reduce generalization and also result in weakly decomposed landscapes, diminishing DADO’s advantage.
>
> > does not explore other practical sources of decomposability…For domains without 3D structural data (e.g., novel peptides or synthetic materials), what alternative methods would you recommend to derive decomposability for DADO?
>
> The reviewer is correct in noting that in addition to using AlphaFold, functional annotation, alignment, or clustering based approaches are also viable ways to obtain an approximate decomposition for protein design. Our results show that DADO is robust to incomplete or slightly inaccurate decompositions (Figs. A6, A7), indicating that such approximate, domain-knowledge-based approaches are viable. We agree that other scientific design problems will call for different, domain-specific approaches, which are outside the scope of this paper; we also highlight ongoing work on data-driven decomposition learning (lines 530-533).
>
> > lacks theoretical bounds on how much decomposition error DADO can tolerate
>
> Prior work (lines 40-45) extensively studies optimization with imperfect predictors. Our decomposed predictors achieve holdout accuracy comparable to dense models (e.g., for AAV, Pearson ~0.92 vs. ~0.91; Fig. A6a), so we rely on this literature rather than introducing new theoretical bounds.

---

### Official Review · Reviewer_59py · 2025-10-31

**Soundness:** 2
**Presentation:** 2
**Contribution:** 2
**Rating:** 2
**Confidence:** 3

**Summary:**

This paper presents an algorithm (DADO) for learning a distribution over a discrete design space to maximize an expected reward, where that reward function can be expressed as a kind of generalized additive model. The authors show favorable performance relative to classical EDA in both synthetic experiments as well as on a task involving the optimization of the score assigned by a protein property predictor developed with the required structure.

**Strengths:**

The authors are focused on an important and challenging problem in scientific design, as the design space in such settings tends to be discrete or combinatorial and therefore is challenging to explore or optimize over. Their focus on reducing the combinatorial complexity of this search problem to one that is significantly more manageable is a worthwhile pursuit and has real applications. The authors motivated the problem well and provided strong justification for an approach such as theirs for more efficiently optimizing the design space. Further, the generalized additive model structure induced by a junction tree that defines the relationship between the design variables and the property being optimized is a unique choice and lends itself to the paper's originality. Improved efficiency for exploring large combinatorial spaces is of great significance to the field, and the authors have proposed a method that performs capably on the considered problems.

**Weaknesses:**

The requirement that the decompositional form be known is a very strong one in practice. Indeed, in the protein property prediction experiments, the authors resorted to a very specifically designed predictor to make use of this decomposition--a design which limits the accuracy of the proposed property predictor. Hence, it is unclear to me whether this method has much utility to the community. It would be helpful if the authors could better profile the impact of the decomposition on the accuracy of the underlying property predictor.

Additionally, I found the the experiments to be rather limited. There is one set of experiments on small synthetically curated systems with shallow junction trees, and another on a protein property prediction task using the same model architecture for each of the four considered proteins. It is difficult from these limited experiments to build good intuition on the method's ability to scale to problem size and complexity. It would be helpful if the authors could better profile or characterize the behavior of their method on a more representative set of problems involving the optimization over a discrete design space. It would also be helpful to establish performance against other baselines for such problems, not just a vanilla EDA.

**Questions:**

How restrictive is the assumed compositionality? From the paper's definition "consider the form f(x) = C1(˜x1) + C2(˜x2), . . . , Cκ(˜xκ), where Ci denotes an arbitrary function on a set of design variables, x˜i," it seems rather general, but the experiments focus on a fairly narrow set of circumstances (i.e., shallow junction trees), so it would be helpful to understand what this definition covers. For example, an arbitrary neural network that takes x as input, transforms it through layers into some d-dimensional embedding, which are then transformed through a linear layer to a scalar output would seem to apply to this definition (in this case, the Ci functions are the individual embedding dimensions which are functions of the x's times the associated weight from the linear layer). Which would indicate to me that this can be applied arbitrarily to neural network predictors, although the experiments suggest this is not in fact the case. Can you clarify?

How well does the method scale to larger sequence lengths compared to those considered in the paper (L < 100)? What about to larger junction trees? Are there important qualitative differences in predictive performance for the underlying property predictors as sequence length increases?

How robust is DADO in settings where the assumed decomposition is incorrect or incomplete?

---

> ### Author Response · Authors · 2025-11-27
> **Response to reviewer 59py**
>
> > It would also be helpful to establish performance against other baselines for such problems, not just a vanilla EDA.
>
> We added two additional baselines, FDA and PPO, and DADO outperforms both on synthetic and protein problems.
>
> > How restrictive is the assumed compositionality?...For example, an arbitrary neural network…
>
> The assumed decomposability is not restrictive for discrete functions: any function can be written in decomposed form, though the resulting graph may be densely-connected. In our protein experiments, DADO operates on decompositions with substantial junction-tree width and multi-variable cliques (Fig. A3). We expanded Sec. A.1 to clarify the equivalent ways decompositions can be expressed.
>
> Regarding arbitrary neural networks: multilayer networks are not decomposable with respect to input variables by default, because embeddings can mix variables in unrestricted ways. A linear layer is decomposable due to linearity, and dense networks can in principle represent a decomposable function, but this is not enforced architecturally. A universal function approximator cannot, by construction, always be decomposed into low-order factors.
>
> > in the protein property prediction experiments, the authors resorted to a very specifically designed predictor to make use of this decomposition--a design which limits the accuracy of the proposed property predictor… It would be helpful if the authors could better profile the impact of the decomposition on the accuracy of the underlying property predictor…How robust is DADO in settings where the assumed decomposition is incorrect or incomplete?
>
> In comparisons with an all-edges (dense) predictor, holdout accuracy is very similar to the decomposed predictors across multiple datasets (e.g., for AAV, Pearson ~0.92 vs. ~0.91; Fig. A6a), suggesting that the use of our decomposed predictive models is not limiting.
>
> We also vary the contact threshold (producing incomplete decompositions; Fig. A6) and randomly add/delete edges (producing slightly incorrect decompositions; Fig. A7). Holdout accuracy remains stable in both cases, indicating that exact identification of the decomposition is not essential.
>
> > The requirement that the decompositional form be known is a very strong one in practice.
>
> For protein problems, we obtain decompositions directly from AlphaFold, and alignment- or clustering-based approaches are also viable (as noted by reviewer 6Cz8). As shown in Figs. A6–A7, performance is robust to decompositions that are slightly incomplete or incorrect.
>
> There is also ongoing work on data-driven decomposition learning (lines 530–533). We do not claim that all problems are decomposable, but when a reasonably accurate decomposition is available—as we demonstrate for multiple protein design tasks—DADO provides a clear optimization advantage.
>
> > How well does the method scale to larger sequence lengths
>
> DADO scales to larger alphabet sizes ($D$) and sequence lengths ($L$). We added results for $D=50,100$ (Fig. A2), $L=200$ (Fig. 2), and $L=400$ (Fig. A1). In all cases, DADO outperforms the decomposition-unaware EDA.
>
> > It is difficult from these limited experiments to build good intuition on the method's ability to scale to problem size and complexity…more representative set of problems
>
> We expanded evaluation with additional synthetic experiments (Figs. A1, A2) and additional protein datasets (Fig. A3). Computational complexity is not a bottleneck on the landscapes tested up to size $20^{400}$ (runtime table in Sec. A.8.7).
>
> Note that each of the seven protein property predictors uses a different architecture depending on its AlphaFold-based decomposition, representing the diversity in objective functions in the real world.
>
> We do not claim that DADO is optimal for all problems; when a function is only weakly decomposable, optimization is difficult for any method. CreiLOV exemplifies such a case, as its junction tree has large nodes (Fig. A3g).
>
> > Are there important qualitative differences in predictive performance for the underlying property predictors as sequence length increases?
>
> Holdout predictive performance does not vary substantially with sequence length (Fig. A6), though the holdout sets necessarily cover only a small fraction of the full design space.
>
> As sequence length grows, the training data cover a smaller proportion of the design space, so predictor accuracy will inevitably degrade on a larger fraction of sequences. This is true for both all-edges (dense) predictors and for decomposed predictors.

---

### Official Review · Reviewer_75LP · 2025-11-01

**Soundness:** 2
**Presentation:** 1
**Contribution:** 1
**Rating:** 6
**Confidence:** 2

**Summary:**

This paper proposes Decomposition-Aware Distributional Optimization (DADO) that leverages decomposability and employs a generative model to guide the search for the target distribution.

DADO is compared to standard Estimation of Distribution Algorithm (EDA), a form of Expectation-Maximization that is not decomposition-aware.

The core of DADO is its use of an objective function decomposed into a junction tree, which enables node-level estimation of value (Q- and V-) functions that represent the choice of variables at each edge and node respectively. These value functions are used to update the search distribution via dynamic programming in the form of message-passing.

DADO outperforms standard EDA on synthetic functions as well as a multi-layer perceptron (MLP)-based predictive model of protein property functions.

**Strengths:**

- DADO is well-motivated and extensively derived
- DADO clearly converges faster than standard EDA
- DADO is a novel, more efficient algorithm than fills a gap in the literature

**Weaknesses:**

- This paper is hard to read due to large amounts of text, in-line math, and few subsections.
- The evaluation is limited to three synthetic functions and four learned protein property functions
- Standard EDA that is unaware of function decomposability can outperform DADO in the absence of ad hoc hyperparameter tuning.
- The GB1 evaluation appears prematurely ended, as the EDA does not appear to converge by the final training iteration

Typo:263 shaing (shaping)

**Questions:**

- How long does it take to run DADO and standard EDA?
- Can DADO still be used for larger alphabet sizes or sequence lengths?
- How would DADO change if positions did not share the same alphabet?

---

> ### Author Response · Authors · 2025-11-27
> **Response to reviewer 75LP**
>
> > This paper is hard to read due to large amounts of text, in-line math, and few subsections.
>
> We appreciate this feedback. For the revision, we reworked the main method section for clarity and added appendix sections with detailed derivations and explanations.
>
> > Can DADO still be used for larger alphabet sizes or sequence lengths?
>
> Yes, DADO supports larger alphabet sizes ($D$) and sequence lengths ($L$). We added results for $D=50,100$ (Fig. A2), $L=200$ (Fig. 2), and $L=400$ (Fig. A1). In all cases, DADO outperforms the decomposition-unaware EDA.
>
> > How would DADO change if positions did not share the same alphabet?
>
> DADO does not assume that positions share the same alphabet. In our synthetic tasks, the per-position alphabets are independent (no relation between alphabets at each position). DADO also handles problems where alphabet sizes differ across positions.
>
> > Standard EDA that is unaware of function decomposability can outperform DADO in the absence of ad hoc hyperparameter tuning.
>
> Our hyperparameter sweep is systematic rather than ad hoc; we apply the same sweep to all methods to ensure a fair comparison for each problem. This avoids giving any method an advantage due to manual tuning or arbitrary hyperparameter choices.
>
> > The evaluation is limited to three synthetic functions and four learned protein property functions.
>
> We expanded evaluation with additional synthetic experiments (Figs. A1–A2) and additional protein experiments (Fig. A3).
>
> > How long does it take to run DADO and standard EDA?
>
> DADO and standard EDA have comparable runtimes (added table in Sec. A.8.7). For example, an $L=100$ run takes ~16 minutes and an $L=400$ run takes ~2 hours. We also report runtimes for larger $D$ and $L$.
>
> > The GB1 evaluation appears prematurely ended, as the EDA does not appear to converge by the final training iteration
>
> We re-ran the GB1 experiments with a larger sample budget, and all baseline methods now converge (Fig. A4). The conclusions about DADO’s superior performance relative to the decomposition-unaware EDA remain unchanged.

---

### Author Response · Authors · 2025-11-27
**General rebuttal and overview of updates**

We thank all reviewers for their thoughtful feedback and for recognizing DADO’s novelty and motivation within the AI-for-science literature. We respond to each individual reviewer’s concerns below, and have updated the manuscript with requested experimental results in the appendix.

**Overview of changes:**

- We substantially reworked the main method section to improve clarity (reviewer 75LP). In the appendix we additionally include a full derivation of DADO (Sec. A.3), an alternate derivation starting from the maximum entropy objective (Sec. A.4), a derivation of the naive EDA (Sec. A.2), and a more in-depth explanation of ways one can write a function decomposition (Sec. A.1).
  - We moved Fig. 4 (ablation of GB1 decomposition graph threshold) to the appendix to make space for a more clear methods section in the main text.
- We added two baselines, FDA and PPO (reviewers dP4b, 59py). PPO is decomposition-unaware; FDA is decomposition-aware but does not use value functions.
  - DADO still outperforms all baselines by a large margin on synthetic problems.
  - On 4 protein problems (Fig. 3), DADO outperforms all baselines (statistically significant).
- Fig. 2 (synthetic experiments) now includes up to L=200 (before only L=100) and results for L=400 are included in the appendix Fig. A1 (reviewers 75LP, 59py). We also performed synthetic experiments with larger state spaces (D=50, D=100) in Sec. A.8.2 / Fig. A2 (reviewers 75LP, 59py, 6Cz8).
- We added experiments on two difficult and highly-decomposable protein datasets, AAV and Gcn4, which include sequences many mutations away from the wild-type (up to 28 and 44 mutations, respectively) to Fig. 3. Compared to the datasets they replace (up to two mutations away from wild-type), these datasets are more uniformly dispersed throughout the design space and are more representative of challenging real-world discrete design problems.
  - Results for the datasets removed are included in the appendix for completeness.
  - To make it easier to visualize performance differences between methods when $f(x)$ is high, we plotted $-\log (c - f(x))$, where $c$ is the highest value on the plot. We include plots without this scaling of the y-axis in the appendix.
- For clarity, we replaced the p-values on the sample mean $f(x)$ AUC with the final iteration sample mean $f(x)$ (still a two-sided t-test over 20 replicates).
- We tested the robustness of our protein property predictors to different decompositions (reviewers 59py, 6Cz8, dP4b). Holdout accuracy remains stable when varying the contact threshold (Fig. A6) or randomly adding/deleting a limited number of edges (Fig. A7), suggesting that identifying the perfect decomposition is not essential.
- We made a table comparing runtimes of DADO and naive EDA for some representative synthetic and protein experiments (reviewer 75LP).

We also want to address concerns about demonstrating DADO on non-protein scientific design problems. As stated in the introduction (lines 45–46), this work focuses on protein design as a proof of concept, even though the method is general. As noted in the discussion (lines 530-533), we agree that studying decomposition in other scientific domains is a compelling direction for future work.

We welcome further discussion and will respond as quickly as possible. We will add main text references to relevant experiments in the appendix and clean up formatting of some of the appendix figures before the camera-ready deadline.

---

### Author Response · Authors · 2025-12-03
**Summary for AC**

Dear AC, while we hope that you’ll have the bandwidth to read the reviews, our responses and updated manuscript, we appreciate the extraordinary circumstances you find yourself in. As such, we here condense these into one place so as to help orient you.

We believe that we’ve addressed all of the key concerns of the reviewers, warranting increased scores for our work.

All four reviewers appeared to be supportive of our method (DADO), its motivation, and its novelty. For example:
- reviewer 6Cz8: “addresses a critical bottleneck in AI-driven scientific design”, “Its contributions are well-aligned with the needs of both machine learning (ML) for optimization and domain sciences (e.g., protein engineering), making it a valuable addition to the field”, “key innovation not seen in prior work”
- reviewer 75LP: “DADO is well-motivated and extensively derived”, “clearly converges faster”, “is a novel, more efficient algorithm than fills a gap in the literature”
- reviewer 59py: “The authors motivated the problem well and provided strong justification for an approach such as theirs”, “Improved efficiency for exploring large combinatorial spaces is of great significance to the field, and the authors have proposed a method that performs capably on the considered problems”
- reviewer dP4b: “the core idea of applying message passing for distributional optimization is novel and an interesting approach to the problem of scientific design”

On the other hand, reviewers were primarily concerned with the following three issues:
- Whether the experiments were sufficiently representative of problems of interest$^{1,2,3,4}$.
> **Our response:** we added synthetic experiments to cover more lengths (up to 400) and alphabet sizes (up to 100), and introduced two more experiments on real-world protein datasets (AAV and Gcn4), which were more difficult ($f(x)$ less smooth) than in the original experiments. Our method (DADO) continued to outperform the baselines (all statistically significant)$^9$.
- Expanding the set of baseline comparisons$^{5,6}$.
> **Our response:** we added two most relevant baselines (FDA and PPO). Our method (DADO) also outperformed these on all experiments in the main text (Figs. 2 and 3; all statistically significant)$^9$. In some auxiliary experiments in the Appendix, this was not so, but we discuss why therein.
- Whether specifying a sufficiently accurate decomposition is feasible for a broad range of real-world design problems$^{7,8}$.
> **Our response:** for each of 5 proteins, we demonstrated that even with a range of decompositions (by varying the contact threshold, and also by randomly adding/deleting a limited number of edges in the decomposition graph), that predictive model holdout accuracy remained stable. This illustrated that identifying the perfect decomposition is not essential for using DADO$^9$.

We also revised the methods section, added run-time comparisons, and answered more specific reviewer requests / questions$^9$. Some reviewers had requested that we demonstrate DADO on non-protein scientific design problems. We first showed through extensive synthetic experiments that DADO leverages structure to improve optimization, and then on protein design problems, an important area in and of itself, that we could deduce an accurate decomposition and that DADO could leverage it for improved optimization. Thus, it is clear that DADO both works, and has demonstrated applicability in at least one scientific domain. Applying it to other domains is beyond the scope of this paper, but we look forward to others trying it out. Thanks for your time and attention; we hope everything is clear!

&nbsp;
### Footnotes
1. reviewer 75LP: "The evaluation is limited to three synthetic functions and four learned protein property functions"
2. reviewer 59py: "It would be helpful if the authors could better profile or characterize the behavior of their method on a more representative set of problems"
3. reviewer 6Cz8: "does not address scalability to larger discrete alphabets"
4. reviewer dP4b: "my main concerns/questions stem from the learned decomposition of the objective function. While there is evidence that DADO works if a good function decomposition exists, it is not clear that (1) such a good function decomposition always exists"
5. reviewer 59py: "It would also be helpful to establish performance against other baselines for such problems"
6. reviewer dP4b: "It seems like a number of baselines are missing from experimental evaluation…[such as] PPO, BO, FDA, MCTS"
7. reviewer 59py: "It would be helpful if the authors could better profile the impact of the decomposition on the accuracy of the underlying property predictor."
8. reviewer 6Cz8: "does not explore other practical sources of decomposability…[what] if you intentionally introduce errors into the junction tree[?]"
9. See “General rebuttal and overview of updates” for details and references to the updated manuscript, and responses for more details.

---

### Meta-Review · Area_Chair_NcHd · 2026-01-07

**Summary:**

This paper proposes Decomposition-Aware Distributional Optimization (DADO) for optimizing distributions over high-dimensional discrete design spaces (e.g., proteins). The key idea is to exploit decomposability of the property predictor via a junction tree: DADO uses a factorized search distribution for sampling, and message passing / dynamic programming over the junction tree to coordinate updates across variables. Reviewers broadly liked the motivation and novelty (message passing for distributional optimization) and found the empirical results promising. The initial weaknesses centered on (i) limited representativeness / scaling of experiments, (ii) missing or incomplete baselines against stronger distributional and black-box methods, and (iii) how feasible and robust it is to specify an accurate decomposition in realistic design settings. The rebuttal adds substantial new experiments (larger lengths and alphabets; additional, harder protein datasets), adds PPO and FDA baselines, includes runtime comparisons, and provides robustness tests showing predictor accuracy is stable under decomposition perturbations.

The paper looks strengthened with remaining risk mainly around generality beyond proteins and reliance on having a usable decomposition. With this, I recommend acceptance.

**Reviewer Concerns:**

Addressed in rebuttal / revision:
* Limited experimental scope / scaling: authors added synthetic experiments with longer sequences (up to 200 in main, 400 in appendix) and larger alphabets (D up to 50/100), and added more challenging real protein datasets (AAV, Gcn4) that are farther from wild-type; DADO continues to outperform baselines.
* Missing baselines: authors added PPO (decomposition-unaware) and FDA (decomposition-aware but lacking value-function component) and report DADO outperforming them on synthetic and protein tasks in main figures, with discussion of exceptions in appendix.
* Runtime questions: authors added runtime table and claim DADO and naive EDA have comparable runtimes on representative settings; they also addressed concerns about prematurely ended GB1 by rerunning with larger budget.
* Robustness to imperfect decompositions: authors tested varying contact thresholds and random edge add/delete; they report holdout accuracy of predictors remains stable, suggesting DADO does not require a perfect decomposition.
* Concern that decomposed predictor may sacrifice accuracy vs dense predictor: authors report comparable holdout accuracy to dense/all-edges predictors on at least some datasets (example numbers given for AAV), and additional tests under perturbed decompositions.
* Data regime / distribution shift concerns: authors added experiments training predictors on restricted subsets (e.g., bottom 50%) and report modest accuracy drops; they argue this is not worse than dense predictors and point to prior literature on optimization with imperfect predictors.
* Scalability to larger alphabets and non-uniform per-position alphabets: authors claim DADO does not require shared alphabets and added synthetic results for larger D.

Still outstanding (or only partially resolved):
* Generality beyond proteins: authors explicitly keep other domains (circuits/molecules/materials) out of scope, so the paper remains a proof-of-concept in proteins + synthetic tasks. This is fine if the claims are scoped accordingly, but it limits “AI-for-science broadly” positioning.
* Theoretical tolerance to decomposition error: robustness is shown empirically, but no formal bound is provided; reviewer 6Cz8’s request is only partially satisfied.
* Dependence on having a reasonable decomposition: authors argue proteins allow this (AlphaFold, etc.) and show robustness to moderate errors, but for domains without structural signals this remains an open practical hurdle (not fatal, but should be framed as a limitation).

**Reviewer Scores:**

* 75LP: original 6. Their main concerns (readability, scale, runtime, GB1 budget) were directly addressed with added experiments, runtime table, and reruns. Estimated final: 7.
* 59py: original 2. They were the most skeptical (assumption that decomposition is known; limited experiments; missing baselines; predictor accuracy tradeoff). Authors addressed each with new baselines, more experiments, and predictor-accuracy/robustness checks. I’d expect a meaningful increase, though maybe still cautious about real-world feasibility. Estimated final: 4-5.
* 6Cz8: original 4. Concerns about larger alphabets, alternative sources of decomposability, and error tolerance were addressed with larger-D synth experiments and robustness tests; theory bounds still missing. Estimated final: 5.
* dP4b: original 4. Concerns about missing black-box baselines, decomposition learning shift, and generality. PPO+FDA were added; predictor robustness and bottom-percentile training were added; generality remains scoped to proteins. Estimated final: 5-6.

---

### Decision · Program_Chairs · 2026-01-26

Accept (Poster)